# Effects of occlusal disharmony on cardiac fibrosis, myocyte apoptosis and myocyte oxidative DNA damage in mice

Yuka Yagisawa[1,2], Kenji Suita[1], Yoshiki Ohnuki[1], Misao Ishikawa[3], Yasumasa Mototani[1], Aiko Ito[2], Ichiro Matsuo[1,4], Yoshio Hayakawa[1,5], Megumi Nariyama[6], Daisuke Umeki[2], Yasutake Saeki[1], Yasuharu Amitani[7], Yoshiki Nakamura[2], Hiroshi Tomonari[2], Satoshi Okumura[1]*

1 Department of Physiology, Tsurumi University School of Dental Medicine, Yokohama, Japan,
2 Department of Orthodontics, Tsurumi University School of Dental Medicine, Yokohama, Japan,
3 Department of Oral Anatomy, Tsurumi University School of Dental Medicine, Yokohama, Japan,
4 Department of Periodontology, Tsurumi University School of Dental Medicine, Yokohama, Japan,
5 Department of Dental Anesthesiology, Tsurumi University School of Dental Medicine, Yokohama, Japan,
6 Department of Pediatric Dentistry, Tsurumi University School of Dental Medicine, Yokohama, Japan,
7 Department of Mathematics, Tsurumi University School of Dental Medicine, Yokohama, Japan

* okumura-s@tsurumi-u.ac.jp

**Data Availability Statement:** All relevant data are within the manuscript and its Supporting Information files.

## Abstract

Occlusal disharmony leads to morphological changes in the hippocampus and osteopenia of the lumbar vertebra and long bones in mice, and causes stress. Various types of stress are associated with increased incidence of cardiovascular disease, but the relationship between occlusal disharmony and cardiovascular disease remain poorly understood. Therefore, in this work, we examined the effects of occlusal disharmony on cardiac homeostasis in bite-opening (BO) mice, in which a 0.7 mm space was introduced by cementing a suitable applicance onto the mandibular incisior. We first examined the effects of BO on the level of serum corticosterone, a key biomarker for stress, and on heart rate variability at 14 days after BO treatment, compared with baseline. BO treatment increased serum corticosterone levels by approximately 3.6-fold and the low frequency/high frequency ratio, an index of sympathetic nervous activity, was significantly increased by approximately 4-fold by the BO treatment. We then examined the effects of BO treatment on cardiac homeostasis in mice treated or not treated with the non-selective β-blocker propranolol for 2 weeks. Cardiac function was significantly decreased in the BO group compared to the control group, but propranolol ameliorated the dysfunction. Cardiac fibrosis, myocyte apoptosis and myocyte oxidative DNA damage were significantly increased in the BO group, but propranolol blocked these changes. The BO-induced cardiac dysfunction was associated with increased phospholamban phosphorylation at threonine-17 and serine-16, as well as inhibition of Akt/mTOR signaling and autophagic flux. These data suggest that occlusal disharmony might affect cardiac homeostasis via alteration of the autonomic nervous system.

**Funding:** This work was supported by Japan Society for the Promotion of Science (JSPS) KAKENHI Grant [20K10305 to Dr. Kenji Suita, 20K10304 to Dr. Yoshiki Ohnuki, 17K17342 to Dr. Daisuke Umeki., 17K11977 to Dr. Megumi Nariyama., 19K24109 to Dr. Aiko Ito, 18K06862, 19H03657 to Dr. Satoshi Okumura]; the MEXT-Supported Program for the Strategic Research Foundation at Private Universities 2015-2019 (S1511018 to Dr. Satoshi Okumura); an Academic Contribution from Pfizer Japan (AC190821 to Dr. Satoshi Okumura); Research Promotion Grant from the Society for Tsurumi University School of Dental Medicine (28006 to Dr. Yuka Yagisawa). The funders had no role in study design, data collection and analysis, decision to publish, or preparation of the manuscript. All authors approve of the contents and agree to coauthorship.

**Competing interests:** SO received funding from Pfizer Japan. There are no patents, products in development or marketed products to declare. This does not alter our adherence to PLoS One policies on sharing data and meterials.

## Introduction

Occlusal disharmony is induced by either loss or incorrect positioning of teeth, causing abnormalities in the force or direction of bite. Patients who suffer from occlusal disharmony occasionally complain about stiffness of the neck or shoulders, fatigue, or psychological stress [1], suggesting that they may suffer from chronic stress. Indeed, previous cross-sectional studies indicate that occlusal disharmony is associated with difficulties in pronunciation and chewing, unsatisfactory facial aesthetics, and emotional turmoil with low self-esteem, low sociality, and poor oral health-related quality of life in adults [2].

Occlusal disharmony in mice, induced by removal of their upper molar teeth, has been shown to elevate plasma corticosteroid level, a marker of stress [3]. Further, occlusal disharmony induced by an occlusal cap splint led to urinary cortisol excretion, as well as bruxism (teeth grinding and jaw clenching), which is associated with emotional stress, in monkeys [4,5]. In addition, an increase of occlusal height induced by cap placement on the incisors resulted in increased levels of serum corticosteroid level and hypothalamic noradrenaline release, together with decreased hippocampal acetylcholine release, in rats [6–8]. These findings suggest that occlusal disharmony might cause stress with activation of sympathetic nerve activity and decreased parasympathetic activity, as well as impaired learning and memory. More recently, occlusal disharmony induced by a tooth height increase (0.5 mm) with composite resin in mice was found to be associated with osteoporosis of the lumbar vertebrate and long bones of the hind limb [9]. All of these results suggest that improvement of occlusal disharmony may yield sustained gains in health and well-being.

Various forms of stress, such as restraint, electrical footshock, cold stress and psychological stress, such as depression and anxiety, are associated with an increased incidence of cardiovascular disease [10–14]. However, the relationship between occlusal disharmony and cardiovascular disease remains poorly understood.

Therefore, the aim of this study was to examine the effects of occlusal disharmony on stress markers, heart rate (HR) control via the autonomic nervous system, systolic cardiac function, histology and signal transduction in the heart, using bite-opening (BO) mice, which have previously been used in research on occlusal disharmony [8,9,15].

## Materials and methods

### Mice and experimental protocol

All experiments were performed on male 12-week-old C57BL/6 mice obtained from CLEA Japan (Tokyo, Japan). Occlusal disharmony in mice was induced by introducing a 0.7-mm BO, by cementing a suitable appliance onto the mandibular incisor under anesthesia with medetomidine (0.03 mg/ml), midazolam (0.4 mg/ml), and butorphanol (0.5 mg/ml), injected intraperitoneally [9,15,16] (**Fig 1A**). Mice were group-housed (approximately 3 mice per cage) at 23°C under a 12–12 light/dark cycle with lights on at 8:00 AM and were divided into four groups: a normal control group (Control), a BO-only treatment group (BO), a propranolol-only treatment group (Pro), and a BO plus propranolol treatment group (BO + Pro) (**Fig 1A**). (±)-Propranolol hydrochloride (#P0884; Sigma, St. Louis MO, USA) was directly dissolved in drinking water (1g/L; freshly prepared every day) [17]. Because the BO mice cannot easily eat the standard pellet food (CE-2: 334.9 kcal/100g; CLEA Japan) but can take paste food, the standard pellet food was changed to paste food three days before the BO treatment in all groups, as in previous studies [9,15]. Body weight (BW), food intake, and water intake were monitored throughout the 2-week experimental period (Control: $n = 6$; BO: $n = 8$; Pro: $n = 6$; BO + Pro: $n = 10$) (**Fig 1B and S1 and S2 Figs of S1 Data**). All animal experiments complied with the

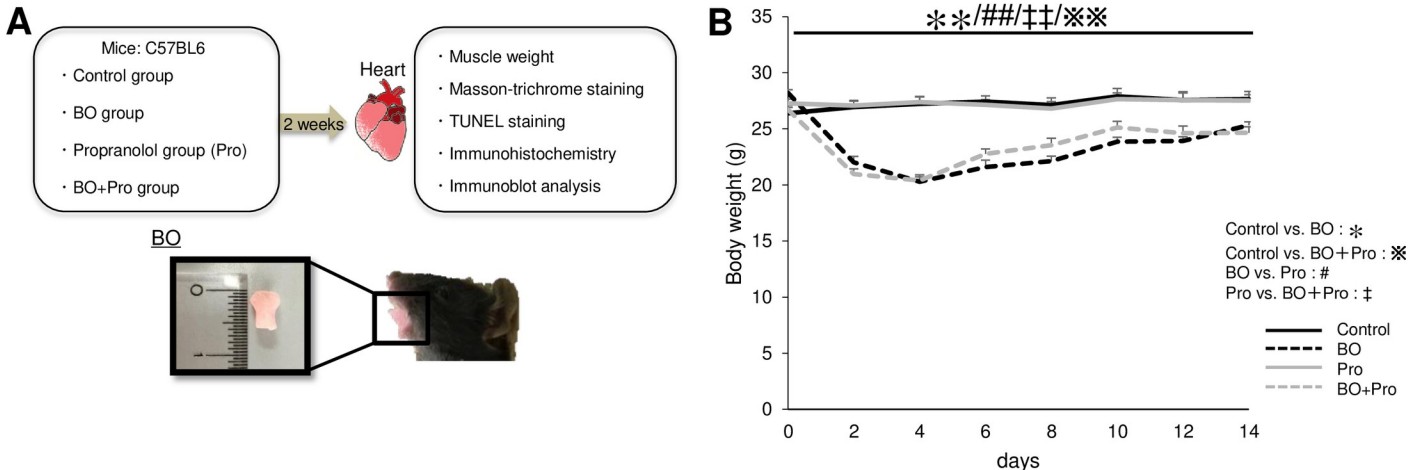

**Fig 1. Experimental procedure and daily body weight. (A)** Male 12-week-old C57BL/6 mice were divided into four groups: a normal control group (Control), a bite-opening (BO)-treated group, a (±)-propranolol hydrochloride (Pro)-treated group, and a BO plus Pro-treated group (BO + Pro). Propranolol was directly dissolved in drinking water (80 mg/kg/day; freshly prepared every day) for 2 weeks. **(B)** Body weight was measured daily for all animals throughout the 2-week experimental period. **P < 0.01 (Control ($n = 6$) vs. BO ($n = 8$)), ※※$P < 0.01$ (Control vs. BO + Pro ($n = 10$)), ##$P < 0.01$ (BO vs. Pro ($n = 6$)), ‡‡$P < 0.01$ (Pro vs. BO + Pro) by two-way repeated-measures ANOVA followed by the Bonferroni *post hoc* test **(S1 Fig of S1 Data)**.

ARRIVE guidelines [18] and were carried out in accordance with the National Institutes of Health guide for the care and use of laboratory animals [19] and institutional guidelines. The experimental protocol was approved by the Animal Care and Use Committee of Tsurumi University (No. 29A041).

## Serum corticosterone measurements

The serum was separated from blood samples collected from the heart of the control ($n = 5$) and BO mice ($n = 5$) under anesthesia at 14 days after the BO treatment. Blood sampling was done in the morning (9:00–10:00AM) and the procedure was completed within 30 s from the time of contact with the mouse [20]. The separated serum samples were frozen at -80°C until measurement. The serum corticosterone levels were determined using a Corticosterone HS EIA kit (#AC-15F1; Immunodiagnostic Systems Ltd., Tyne & Wear, UK), according to the manufacturer's instructions.

## Physiological experiments

Mice were anesthetized via a mask with isoflurane (1.0–1.5% v/v) at room temperature to maintain the lightest anesthesia possible and echocardiographic measurements (Control: $n = 10$; BO: $n = 7$; Pro: $n = 5$; BO + Pro: $n = 7$) were performed by means of ultrasonography (TUS-A300, Toshiba, Tokyo, Japan) at 14 days after the BO treatment [21]. After the completion of echocardiographic measurement, mice were anesthetized via a mask with isoflurane (1.0–1.5% v/v) at room temperature and killed by cervical dislocation [22,23]. The heart was excised, rinsed thoroughly in phosphate-buffered saline to eliminate circulating blood in tissue, blotted on filter paper and weighed. The cardiac muscle mass (CMM; mg), the ratio of CMM to tibial length ratio (mm) and the ratio of CMM to BW (g) were used as an indexes of muscle growth. For the immunoblotting analysis, the excised heart tissue was immediately frozen in liquid nitrogen and stored at -80°C until the preparation of crude protein homogenate. For the histologic analysis, the excised heart tissue was immediately frozen in liquid nitrogen

with Tissue-Tek OCT compound (Sakura Finetek, Torrance, CA, USA) and stored at -80˚C until sectioning.

## Electrocardiogram acquisition and analysis

Mice were anesthetized with intraperitoneal medetomidine (0.03 mg/ml), midazolam (0.4 mg/ml), and butorphanol (0.5 mg/ml). Then, an abdominal midline incision was made on the ventral surface, and a transmitter (F20-EET; Data Sciences International, St. Paul, MN, USA) was implanted into the mice ($n$ = 5) at 14 days before the BO treatment. Electrocardiogram (ECG) signals from the telemetric units in freely moving mice in plastic cages were recorded on an under-cage receiver (Data Sciences International, St. Paul, MN, USA), digitized at a sample rate of 2 kHz, and fed into a microcomputer-based data acquisition system (Power Lab System, AD Instruments, Milford, MA, USA). ECG data were recorded for 24 h at 1 day before the BO treatment to obtain the baseline and at 1, 7 and 14 days after the BO treatment (**Fig 2A**) [24,25]. ECG signals processing was performed with Chart v5.0 software and heart rate variability (HRV) analysis was done with the HRV plug-in for Chart v5.0 (AD Instruments). This software detects R waves from all ECG leads after passing the signals through a filter that eliminates noise and applying an algorithm that detects ECG fiducial points. All R-R interval data were screened on the computer to confirm the sinus origin of the rhythm. We evaluated the ratio of low frequency (LF; 0.4–1.5 Hz) and high frequency (HF; 1.5–4.0 Hz) as a marker of sympathetic activity, and normalized HF (nHF) as a marker of parasympathetic activity to examine the effects of BO treatment [24,25]. We also evaluated the standard deviation of normal R-R intervals (SDNN), which is a measure of total autonomic instability [24,25].

## Evaluation of fibrosis

Cross sections (10 μm) (Control: $n$ = 6; BO: $n$ = 6; Pro: $n$ = 6; BO + Pro: $n$ = 6) were cut with a cryostat (CM1900, Leica Microsystems, Nussloch, Germany) at -20˚C. The sections were air-dried and fixed with 4% paraformaldehyde (v/v) in 0.1M phosphate-buffered saline (pH 7.5) [22,26,27].

Interstitial fibrosis was evaluated by Masson-trichrome staining using the Accustatin Trichrome Stain Kit (#HT15-1KT; Sigma) in accordance with the manufacturer's protocol [26,27]. Interstitial fibrotic regions were quantified using image software analysis (Image J 1.45) of the percentage of blue area in the Masson-trichrome sections [21,26,27].

## Evaluation of apoptosis

Apoptosis was determined by terminal deoxyribonucleotidyl transferase (TdT)-mediated biotin-16-deoxyuridine triphosphate (dUTP) nick-end labeling (TUNEL) staining using the Apoptosis *in situ* Detection Kit (#293–71501; Wako, Osaka, Japan). TUNEL-positive nuclei per field of view were manually counted in six sections from the four groups (Control; $n$ = 6, BO; $n$ = 6, Pro; $n$ = 6, BO + Pro; $n$ = 6) over a microscopic field of 20 x, averaged and expressed as the ratio of TUNEL-positive nuclei (%) [21,26–28]. Limiting the counting of total nuclei and TUNEL-positive nuclei to areas with a true cross section of myocytes made it possible to selectively count only those nuclei that were clearly located within myocytes.

## Western blotting

Cardiac muscle excised from the mice was homogenized in a Polytron (Kinematica AG, Lucerne, Switzerland) in ice-cold RIPA buffer (Thermo Fisher Scientific, Waltham, MA, USA: 25 mM Tris-HCl (pH 7.6), 150 mM NaCl, 1% NP-40, 1% sodium deoxycholate, 0.1% SDS)

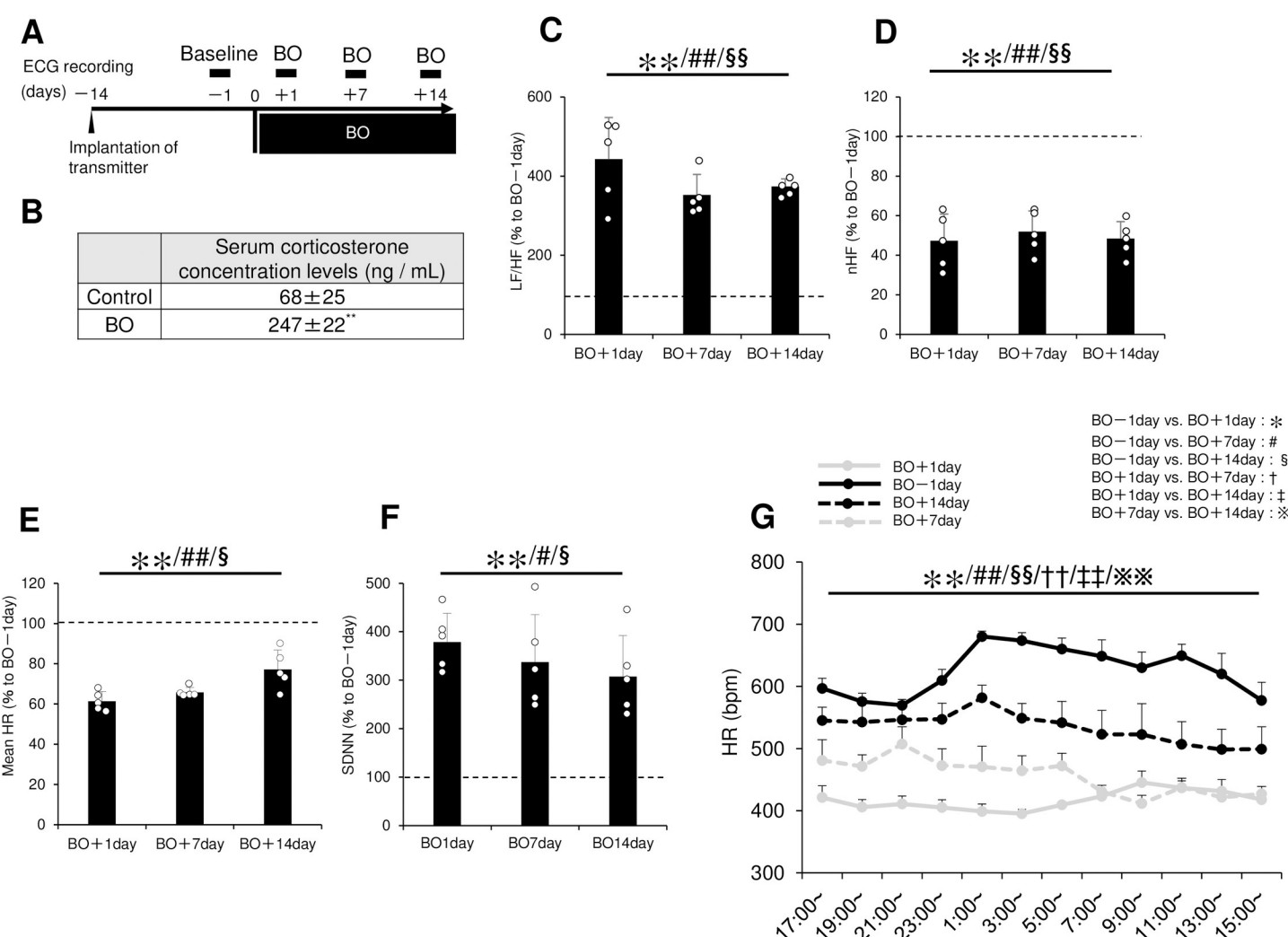

**Fig 2. Effects of BO on serum corticosterone levels, LF/HF, nHF, mean HR and HRV. (A)** ECG was recorded for 24 h at 1 day before the BO treatment (Baseline; BO-1day) and at 1, 7 and 14 days after the BO treatment (BO). **(B)** Serum corticosterone level was significantly increased by the BO treatment for 14 days, compared to the control group. **P < 0.01 by Student *t*-test **(S3A Fig of S1 Data)**. **(C)** LF/HF, an index of the sympathetic nervous activity, was significantly greater at all time points in the BO group, compared to the baseline. **P < 0.01 (BO-1day vs. BO+1day), ##P < 0.01 (BO–1day vs. BO+7day), §§P < 0.01 (BO–1day vs. BO+14day) by one-way repeated-measures ANOVA followed by the Bonferroni *post hoc* test **(S3B Fig of S1 Data)**. **(D)** nHF, an index of parasympathetic activity, was significantly smaller at all time points in the BO group, compared to the baseline. **P < 0.01 (BO-1day vs. BO+1day), ##P < 0.01 (BO–1day vs. BO+7day), §§P < 0.01 (BO–1day vs. BO+14day) by one-way repeated-measures ANOVA followed by the Bonferroni *post hoc* test **(S3C Fig of S1 Data)**. **(E)** Mean HR was significantly smaller at all time points in the BO group, compared to the baseline. **P < 0.01 (BO-1day vs. BO+1day), ##P < 0.01 (BO–1day vs. BO+7day), §§P < 0.05 (BO–1day vs. BO+14day) by one-way repeated-measures ANOVA followed by the Bonferroni *post hoc* test **(S3D Fig of S1 Data)**. **(F)** SDNN was significantly greater at all time points in the BO group, compared to the baseline. **P < 0.01 (BO-1day vs. BO+1day), ##P < 0.01 (BO–1day vs. BO+7day), §§P < 0.05 (BO–1day vs. BO+14day) by one-way repeated-measures ANOVA followed by the Bonferroni *post hoc* test **(S4A Fig of S1 Data)**. **(G)** Time course changes in heart rate during 24 h of ECG measurement at 1 day before and 1, 7 and 14 days after BO treatment. **P < 0.01 (BO-1day vs. BO+1day), ##P < 0.01 (BO–1day vs. BO+7day), §§P < 0.01 (BO–1day vs. BO+14day), ††P < 0.01 (BO+1day vs. BO+7day), ‡‡P < 0.01 (BO+1day vs. BO+14day) and ※※P < 0.01 (BO+7day vs. BO+14day) by two-way repeated-measures ANOVA followed by the Bonferroni *post hoc* test **(S4B of S1 Data)**.

without addition of protein inhibitors [29], and the homogenate was centrifuged at 13,000 $_x g$ for 10 min at 4°C. The supernatant was collected and the protein concentration was measured using a DC protein assay kit (Bio-Rad, Hercules, CA, USA). Equal amounts of protein (5 μg) (Control; *n* = 6, BO; *n* = 6, Pro; *n* = 6, BO + Pro; *n* = 6) were subjected to 12.5% SDS-polyacrylamide gel electrophoresis and blotted onto 0.2 mm PVDF membrane (Millipore, Billerica, MA, USA).

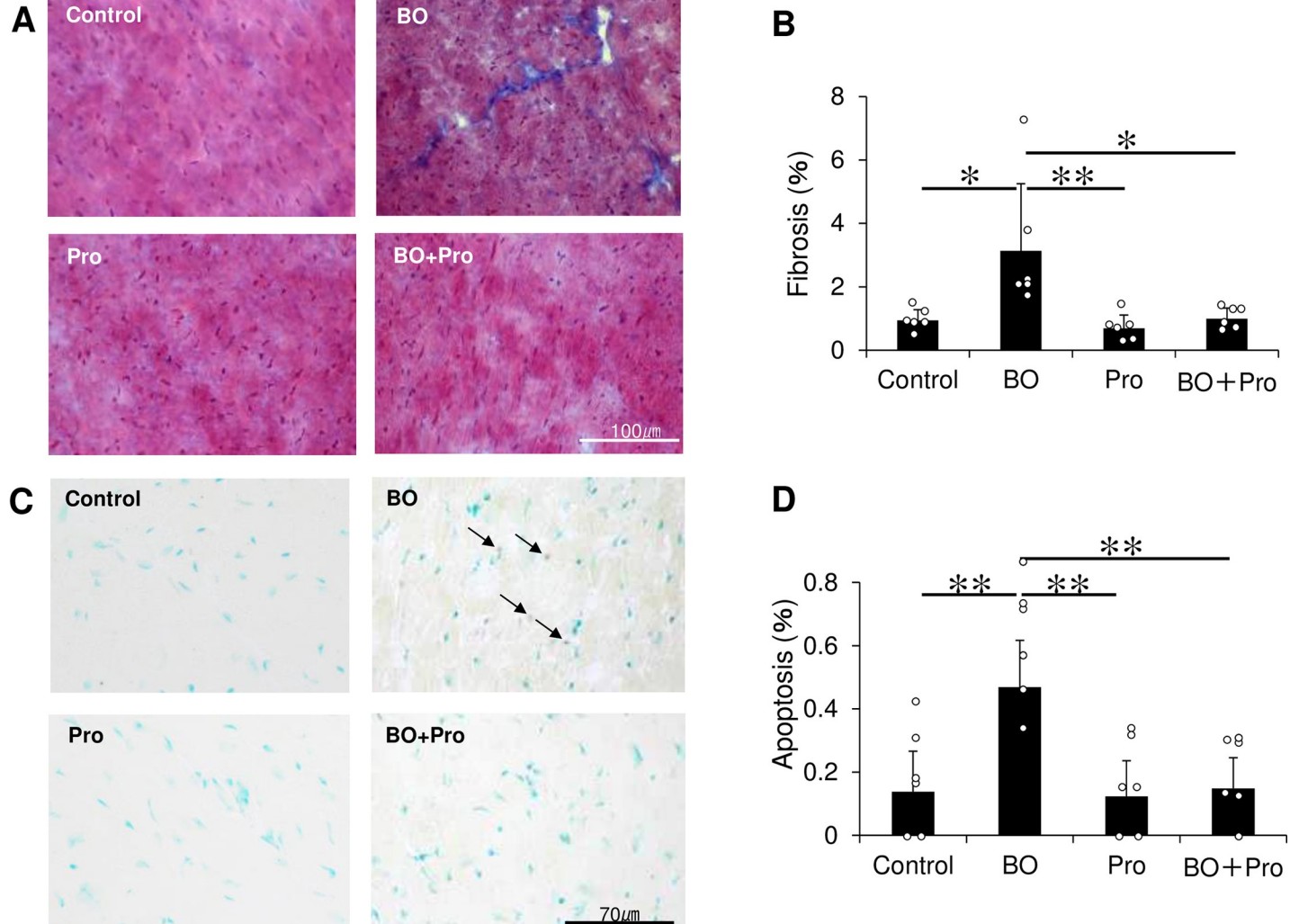

**Fig 3. Effects of BO on fibrosis and apoptosis in the heart.** (A) Representative images of Masson-trichrome-stained sections of cardiac muscle in the Control (*upper left*), BO (*upper right*), Pro (*lower left*) and BO + Pro (*lower right*) groups. (B) The area of fibrosis was significantly increased in the BO group, but this increase was blocked in the BO + Pro group. *$P < 0.05$ or **$P < 0.01$ by one-way repeated-measures ANOVA followed by the Tukey-Kramer *post hoc* test (**S5A Fig of S1 Data**). (C) TUNEL-positive nuclei (black arrows) in representative TUNEL-stained sections were counted in cardiac muscle in the Control (*upper left*), BO (*upper right*), Pro (*lower left*) and BO + Pro (*lower right*) groups. (D) The number of TUNEL-positive nuclei was significantly increased in the BO group, but this increase was blocked in the BO + Pro group. **$P < 0.01$ by one-way ANOVA followed by the Tukey-Kramer *post hoc* test (**S5B Fig of S1 Data**). Data show means ± SD and scattered dots show individual data.

Western blotting was conducted with commercially available antibodies [21,28,30,31]. Primary antibodies directed against the following proteins were purchased from the indicated sources: Akt (1:1000, #9272) [26], phospho-Akt (1:1000, Ser-473, #9721) [26], CaMKII (1:1000, #3362) [32], phospho-CaMKII (1:1000, Thr-286, #3361) [32], BAX (1:1000, #2772) [21], LC3 (1:1000, #12741) [21], Bcl-2 (1:1000, #3498) [28], phospho-mTOR (1:1000, Ser-2448, #5536; Ser-2481, #2974) [26], mTOR (1:1000, #2972) [26] and RIP3 (1:1000, #95702) [33] from Cell Signaling Technology (Boston, MA, USA), p62 (#PM045) from MBL (Nagoya, Japan), and GAPDH (sc-25778) [26] from Santa Cruz Biotechnology (Santa Cruz, CA, USA), phosphorylated phospholamban (PLN) (1:5000, phospho-Ser-16, #A010-12; 1:1000, phospho-Thr-17, #A010-13) [21] and PLN (1:2000, #A010-14) [21] from Badrilla (Leeds, UK). Horseradish peroxidase-conjugated anti-rabbit (1:1000, #NA934) or anti-mouse IgG (#NA931) antibodies

[26] purchased from GB Healthcare were used as secondary antibodies. The primary and secondary antibodies were diluted in Tris-buffered saline (pH 7.6) with 0.1% Tween 20 and 5% bovine serum albumin. Protein oxidation was measured using the OxiSelect™Protein Carbonyl Immunoblot Kit (#STA-308; Cell Biolabs, Inc. San Diego, CA, USA) according to the manufacturer's instructions [34,35]. The blots were visualized with enhanced chemiluminescence solution (ECL: Prime Western Blotting Detection Reagent, GE Healthcare, Piscataway, NJ, USA) and scanned with a densitometer (LAS-1000, Fuji Photo Film, Tokyo, Japan). The reason why there are different numbers of samples in different western blotting figures (**Figs 4–6, S6, S8 and S9 Figs of S1 Data**) is that we excluded outliers (extremely low or high values, compare to others in the same group).

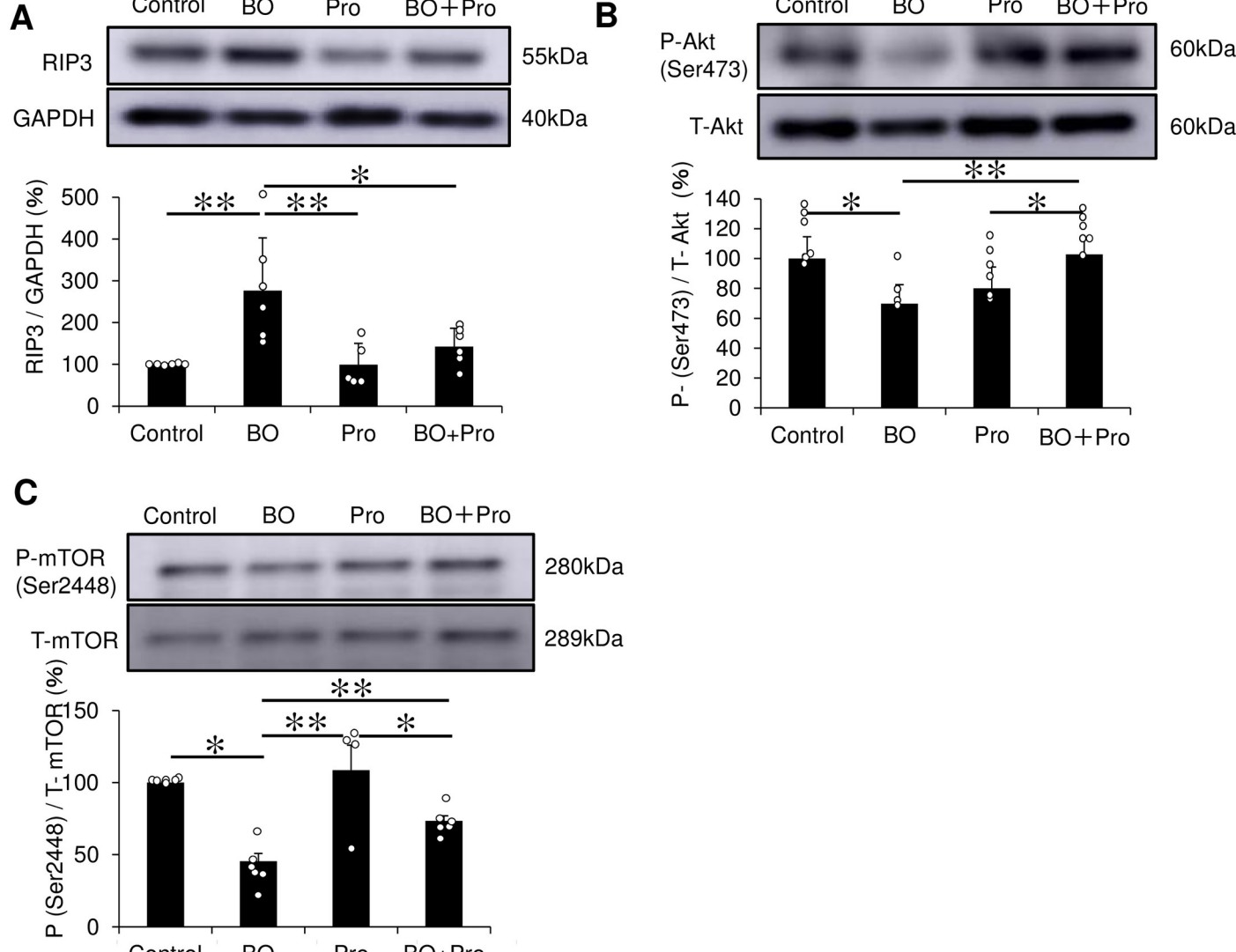

**Fig 4. Effects of BO on RIP3 and Akt/mTOR signaling in the heart. (A)** Expression of RIP3, a key mediator of necroptosis, was significantly increased in the BO group, but this increase was blocked in the BO + Pro group. $^*P < 0.05$ or $^{**}P < 0.01$ by one-way ANOVA followed by the Tukey-Kramer *post hoc* test (**S7A Fig of S1 Data**). **(B)** Akt phosphorylation at Ser 473 was significantly decreased in the BO group, but this decrease was blocked in the BO + Pro group. $^*P < 0.05$ or $^{**}P < 0.01$ by one-way ANOVA followed by the Tukey-Kramer *post hoc* test (**S7B Fig of S1 Data**). **(C)** mTOR phosphorylation at Ser 2448, a specific marker of mTORC1 formation, was significantly decreased in the BO group, but this decrease was blocked in the BO + Pro group. $^*P < 0.05$ or $^{**}P < 0.01$ by one-way ANOVA followed by the Tukey-Kramer *post hoc* test (**S7C Fig of S1 Data**). Data show means ± SD and scattered dots show individual data. Full-size images of immunoblots are presented in S1 Data of supporting information.

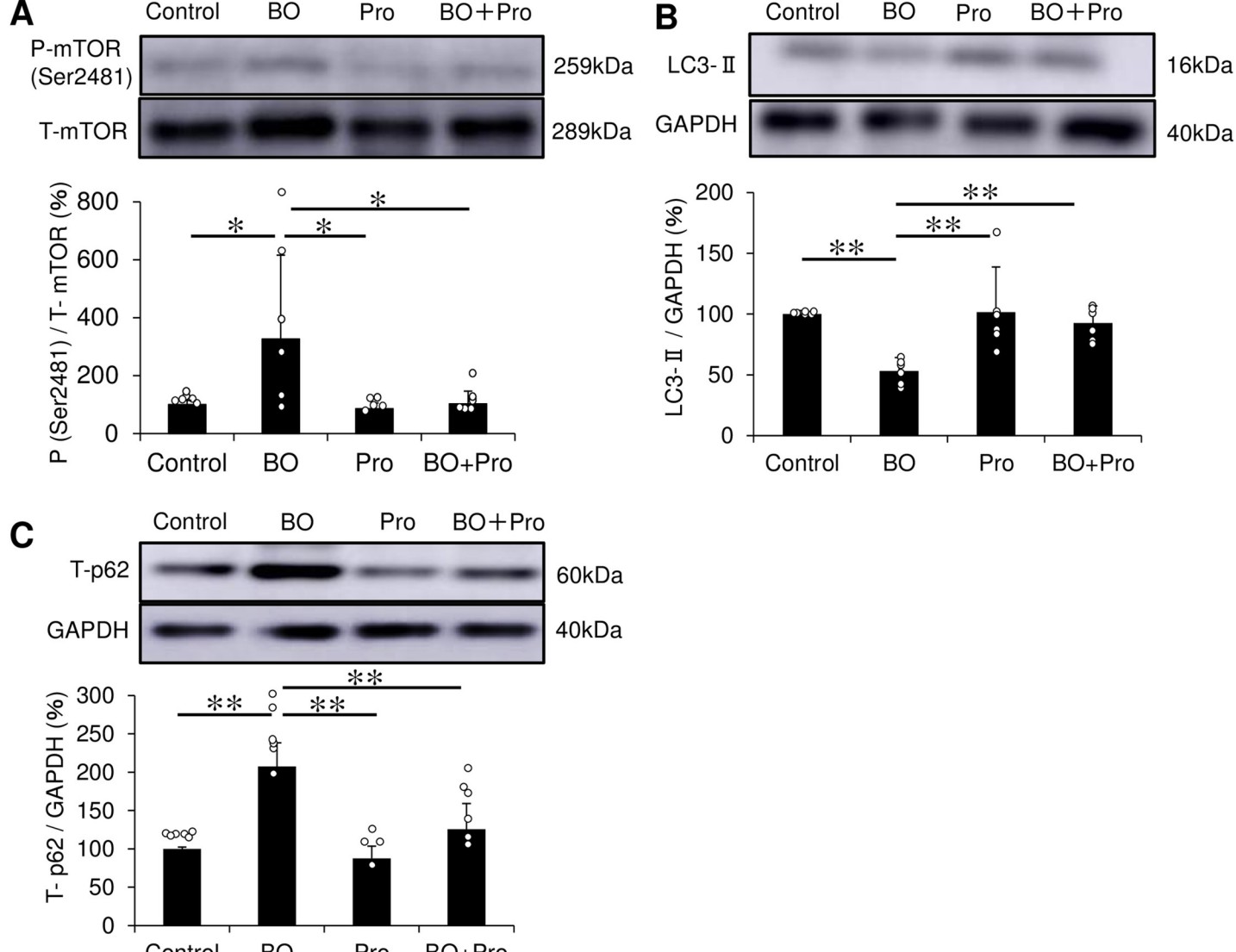

**Fig 5. Effects of BO on mTORC2 phosphorylation, LC3 and p62 expression in cardiac muscle.** (**A**) mTOR phosphorylation at Ser 2481, a specific marker of mTORC2 formation, was significantly increased in the BO group, but this increase was blocked in the BO + Pro group. *$P < 0.05$ by one-way ANOVA followed by the Tukey-Kramer post hoc test (**S10A Fig of S1 Data**). (**B**) Expression of LC3-II, an autophagosome marker, was significantly decreased in the BO group, but this decrease was blocked in the BO + Pro group. **$P < 0.01$ by one-way ANOVA followed by Tukey-Kramer *post hoc* test (**S10B Fig of S1 Data**). (**C**) p62 expression, which correlates inversely with autophagic degradative activity, i.e., autophagic flux, was significantly increased in the BO group, but this increase was blocked in the BO + Pro group. **$P < 0.01$ by one-way ANOVA followed by the Tukey-Kramer *post hoc* test (**S10C Fig of S1 Data**). Data show means ± SD and scattered dots show individual data. Full-size images of immunoblots are presented in S1 Data of supporting information.

## Immunostaining

Oxidative DNA damage in the myocardium was evaluated by immunostaining for 8-hydroxy-2'-deoxyguanosine (8-OHdG) using the Vector M.O.M Immunodetection system (#PK-2200, Vector Laboratories, Inc. Burlingame, CA, USA) [36,37]. Cross sections (Control; $n = 6$, BO; $n = 6$, Pro; $n = 6$, BO + Pro; $n = 6$), were cut with a cryostat at -20˚C at 10 μm, air-dried and fixed with 4% paraformaldehyde (v/v) in TBS-T for 5 min at room temperature. Antigen retrieval was achieved with 0.1% citrate plus 1% Triton X-100 for 30 min at room temperature, then the sections were washed with TBS-T, incubated with 0.3% horse serum in TBS-T for 1 h

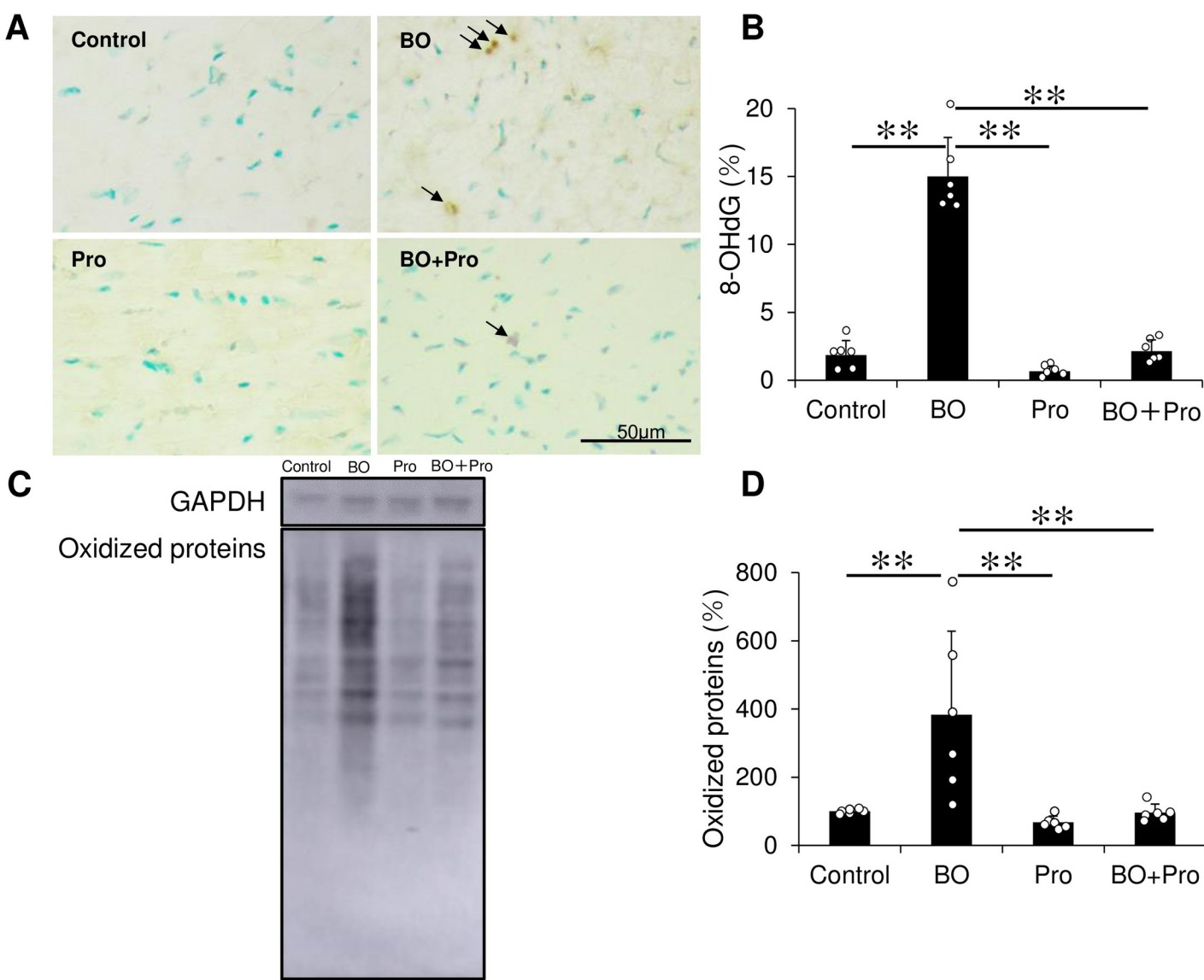

**Fig 6. Effects of BO on oxidative stress in cardiac muscle.** (A) Representative images of immunohistochemical detection of oxidative DNA damage (8-OHdG) in cardiac muscle in the Control (*upper left*), BO (*upper right*), Pro (*lower left*) and BO + Pro (*lower right*) groups. (B) 8-OHdG-positive nuclei was significantly increased in the BO group, but this increase was blocked in the BO + Pro group. $^{**}P < 0.01$ by one-way ANOVA followed by the Tukey-Kramer post hoc test **(S11A Fig of S1 Data)**. (C) Representative SDS-PAGE of oxidized proteins in cardiac muscle homogenate prepared from Control (*lane 1*), BO (*lane 2*), Pro (*lane 3*) and BO + Pro (*lane 4*) groups using the OxiSelect$^{TM}$Protein Carbonyl Immunoblot Kit. (D) Oxidized proteins were significantly increased in the BO group, but this increase was blocked in the BO + Pro group. $^{**}P < 0.01$ by one-way ANOVA followed by the Tukey-Kramer post hoc test **(S11B Fig of S1 Data)**. Data expressed as means ± SD and scattered dots show individual data.

at room temperature, and blocked with M.O.M. blocking reagents (Vector Laboratories, Burlingame, CA, USA) overnight at 4˚C. For the positive control, sections were incubated with 0.3% $H_2O_2$ in TBS-T before the anti-8-OHdG antibody treatment. The sections were incubated with anti-8-OHdG antibody (8.3 μg/ml in M.O.M. Dilute; clone N45.1 monoclonal antibody; Japan Institute for the Control of Aging, Shizuoka, Japan) overnight at 4˚C in a humidified chamber, and then incubated with 0.3% $H_2O_2$ in 0.3% horse serum for 1 h at room temperature to inactivate endogenous peroxidase, rinsed with TBS-T, incubated with anti-mouse IgG in M.O.M. Diluent, and processed with an ABC kit (Vector Laboratories, Inc.

**Table 1. Heart size and cardiac function.**

|  | Control (n) | Bite opening (n) | Propranolol (n) | Bite opening Propranolol (n) |
|---|---|---|---|---|
| **Body weight (mg)** | 28 ± 1.7 (6) | 25 ± 1.7 (8) | 28 ± 1.3 (6) | 25 ± 1.0 (10) |
| **CMM (mg)** | 133 ± 26.9 (6) | 119 ± 15.2 (8) | 136 ± 17.2 (6) | 123 ± 14.3 (10) |
| **CMM/tibia length (mg/mm)** | 6.6 ± 1.2 (6) | 5.9 ± 0.8 (8) | 6.7 ± 0.7 (6) | 6.1 ± 0.7 (10) |
| **CMM/body weight (mg/g)** | 4.8 ± 0.9 (6) | 4.7 ± 0.5 (8) | 4.9 ± 0.5 (6) | 5.0 ± 0.5 (10) |
| **LVEDD (mm)** | 4.3 ± 0.3 (10) | 4.0 ± 0.3 (7) | 4.1 ± 0.2 (5) | 4.3 ± 0.2 (7) |
| **LVESD (mm)** | 2.8 ± 0.1 (10) | 2.8 ± 0.2 (7) | 2.9 ± 0.1 (5) | 2.9 ± 0.2 (7) |
| **LVEF (%)** | 70 ± 2.0 (10) | 63 ± 0.9 (7)** | 61 ± 1.8 (5)** | 69 ± 2.0 (7) |
| **%FS** | 35 ± 1.7 (10) | 30 ± 0.6 (7)** | 28 ± 1.1 (5)** | 33 ± 1.5 (7) |

Data are mean ± SD, CMM; cardiac muscle mass

LVEDD; left ventricular end-diastolic diameter

LVESD; left ventricular end-systolic diameter

LVEF; Left ventricular ejection fraction

%FS; % fractional shortening

Burlingame, CA, USA). We calculated the ratio of 8-OHdG nuclei with oxidative DNA damage (stained dark blown) per total cell numbers.

## Method validation

The procedures used in this study were similar to those used in our previous work: echocardiography [21,38], HRV analysis [24,25], Masson-trichrome staining and TUNEL staining [21,26], western blotting [22,39] and immunostaining [22,26], and each method was validated for reliability and reproducibility for each procedure.

## Statistical analysis

Data are expressed as means ± SD. Comparison of data was performed using a Student's *t*-test for 2 groups (**Fig 2B**), one-way repeated-measures analysis of variance (ANOVA) followed by the Bonferroni *post hoc* test (**Fig 2C–2F**), two-way repeated-measures ANOVA followed by the Bonferroni *post hoc* test (**Fig 2G, S2A, S2C Fig of S1 Data**) or one-way ANOVA followed by the Tukey-Kramer *post hoc* test for 3 or more groups (**Figs 3B, 3D, 4A–4C, 5A–5C, 6B and 6D, S6A-S6B, S8, S9A-S9B Figs of S1 Data, Table 1**). Normality assumption was verified using the Shapiro-Wilk test for all data.

The total sample size of animals required for statistical validity was calculated for an α risk of 0.05 and a statistical power (1-β) of 0.8 [40]. Analyses were performed with PASW statistics 18 (SPSS Inc., Chicago, IL, USA) except for the sample size estimation, which was performed by G*Power version 3.1. (program, concept and design by Franz, Universitat Kiel, Germany; freely available Windows application software) [41]. The criterion of significance was taken as $P < 0.05$.

## Results

### Effects of BO on body weight

We monitored the BW of the four groups daily (**Fig 1B**). BW of the Control and Pro groups was similar and showed no significant change during the experimental period. Conversely, BW of the BO and BO + Pro groups gradually decreased and reached a minimum at 4 days after the BO treatment in accordance with previous findings [9,42] (**Fig 1B, S1 Fig of S1**

**Data**). After that, the BW of the BO and BO + Pro groups gradually increased, but did not reach the preoperative level during the experimental period at 14 days after BO treatment (**Table 1**, S12A Fig of **S1 Data**).

### Effects of BO on the consumption of food and drinking water

We monitored the daily consumption of pellet food and water per mouse, measured as an average of group-housed mice in each cage (approximately 3), during the 2-week experimental period. Consumption levels of food (**S2A and S2B Fig of S1 Data**) and water (**S2C and S2D Fig of S1 Data**) in the Control and Pro groups were similar and did not show significant changes during the experimental period. The BO and the BO + Pro groups might have some difficulty eating, and the consumption of food and water was minimum at 1 day after the BO treatment. However, consumption recovered gradually to preoperative levels within 4 days and no significant difference was observed among the four groups at 2 weeks (**S2A and S2B Fig of S1 Data**). Changes in the consumption of water showed a similar tendency to those of food (**S2C and S2D Fig of S1 Data**).

### Effects of BO on serum corticosterone levels

Comparison of the levels of serum corticosterone level, a key biomarker for stress [6,9], in the control and BO mice at 14 days after the BO treatment revealed a significantly increase of approximately 3.6-fold at 14 days after BO treatment ($n$ = 5 each) (**Fig 2B**, S3A Fig of **S1 Data**). These data suggest that the mice are stressed at 14 days after the BO treatment.

### Effects of BO on LF/HF and nHF

To evaluate changes in autonomic nervous activity, we carried out HRV analysis and compared the ratio of LF to HF (LF/HF), an index of the sympathetic nervous activity [25], at 1 day before (BO-1 day) and at 1 (BO+1 day), 7 (BO+7day) and 14 days (BO+14day) after the BO treatment. LF/HF was significantly greater than baseline at all time points ($P < 0.01$ by one-way repeated-measures ANOVA followed by Bonferroni *post hoc* test, $n$ = 5 each) (**Fig 2C**, **S3B Fig of S1 Data**). HF power was normalized to account for differences in total power (nHF), and nHF was examined as an index of parasympathetic activity [25]. After BO, nHF was significantly decreased from baseline at all time points ($P < 0.01$ by one-way repeated-measures ANOVA followed by the Bonferroni *post hoc* test, $n$ = 5 each) (**Fig 2D**, S3C Fig of **S1 Data**).

These data suggest that sympathetic nerve activity was increased but parasympathetic activity was decreased after BO treatment.

### Effects of BO on HR

To evaluate changes of HR, we examined the mean (**Fig 2E**, S3D Fig of **S1 Data**) and circadian variation of HR (**Fig 2G**, S4B Fig of **S1 Data**) at 1 day before the BO treatment to obtain the baseline (BO-1day) and at 1 (BO+1 day), 7 (BO+7day) and 14 days (BO+14day) after the BO treatment.

Mean HR was unexpectedly but significantly decreased at 1, 7 and 14 days after the treatment of BO, compared to the baseline (BO-1day vs. BO+1day, $P = 3.2 \times 10^{-2}$; BO-1day vs BO+7day, $P = 3.6 \times 10^{-5}$; BO-1day vs. BO+14day, $P = 3.6 \times 10^{-2}$ by one- way repeated-measures ANOVA followed by the Bonferroni *post hoc* test, $n$ = 5 each) (**Fig 2E**, S3D Fig of **S1 Data**).

We also examined the circadian variation of HR and found that it was also decreased by the BO treatment (BO-1day vs. BO+1day, $P = 1.6 \times 10^{-56}$; BO-1day vs BO+7day, $P = 1.2 \times 10^{-44}$;

BO-1day vs. BO+14day, $P = 7.1$ x $10^{-19}$ by two-way repeated-measures ANOVA followed by the Bonferroni *post hoc* test, $n = 5$ each) (**Fig 2G, S4B Fig of S1 Data**).

These data suggest that BO treatment alters the control of HR via the autonomic nervous system.

## Effects of BO on SDNN

Because the above findings indicated a difference in HR regulation after the treatment of BO, we examined SDNN, which is a measure of total autonomic instability [24,25].

SDNN was significantly increased at all time points after the BO treatment, compared to the baseline (BO-1day vs. BO+1day, $P = 2.8$ x $10^{-3}$; BO-1day vs BO+7day, $P = 3.3$ x $10^{-2}$; BO-1day vs. BO+14day, $P = 3.2$ x $10^{-2}$ by one-way repeated-measures ANOVA followed by the Bonferroni *post hoc* test, $n = 5$ each) (**Fig 2F, S4A Fig of S1 Data**), suggesting that autonomic control of the HR was altered after the BO treatment.

## Effects of BO on heart size and cardiac function

We examined the effects of BO on heart size in terms of CMM (mg), CMM per tibial length ratio (mg/mm) and CMM per body weight ratio (mg/g) (**Table 1, S12 Fig of S1 Data**), and they were similar in all four groups. However, we cannot rule out the possibility that the statistical power was insufficient to detect BO-mediated cardiac hypertrophy as the total sample size in these cases were not sufficient to provide an α risk of 0.05 and statistical power (1-β) of 0.8 (**S12 Fig of S1 Data**).

We also conducted echocardiography (**Table 1, S13 Fig of S1 Data**) to evaluate cardiac function in terms of left ventricular ejection fraction (LVEF) and fractional shortening (%FS). Both parameters were significantly decreased in the BO and Pro groups compared to the control. However, no significant changes of LVEF and %FS were observed in the BO + Pro group compared to the control. Also, no significant differences of left ventricular end-diastolic (LVEDD) and left ventricular end-systolic diameter (LVESD) were observed, although we cannot rule out the possibility that the statistical power was insufficient to detect BO-mediated effects on LVEDD and LVESD due to the limited total sample sizes (**S13 Fig of S1 Data**).

These data suggest that BO treatment decreased cardiac function without altering the weight of cardiac muscle.

## Effects of BO on cardiac fibrosis and apoptosis

We examined the effects of BO treatment on fibrosis in cardiac muscle by means of Masson-trichrome staining (**Fig 3A**). BO treatment significantly increased the area of fibrosis in cardiac muscle (Control ($n = 6$) vs. BO ($n = 6$); $0.9 \pm 0.3$ vs. $3.1 \pm 2.1\%$, $P = 1.2$ x $10^{-2}$ by one-way ANOVA followed by the Tukey-Kramer *post hoc* test). Propranolol alone did not alter the area of fibrosis, but it blocked the BO-induced increase of fibrosis (BO ($n = 6$) vs. BO + Pro ($n = 6$); $3.1 \pm 2.1$ vs. $1.0 \pm 0.3\%$, $P = 1.5$ x $10^{-2}$ by one-way ANOVA followed by the Tukey-Kramer *post hoc* test) (**Fig 3B, S5A Fig of S1 Data**).

We also examined the effects of BO treatment on myocyte apoptosis in cardiac muscle by means of TUNEL staining (**Fig 3C**). Myocyte apoptosis in cardiac muscle was significantly increased by BO treatment (Control ($n = 6$) vs. BO ($n = 6$); $0.14 \pm 0.13$ vs. $0.47 \pm 0.15\%$, $P = 7.9$ x $10^{-4}$ by one-way ANOVA followed by the Tukey-Kramer *post hoc* test). Propranolol alone ($n = 6$) had no effect on the number of TUNEL- positive cardiac myocytes, but it blocked the increase of TUNEL-positive cardiac myocytes induced by BO treatment (BO ($n = 6$) vs. BO + Pro ($n = 6$); $0.47 \pm 0.15$ vs. $0.15 \pm 0.10\%$, $P = 1.1$ x $10^{-3}$ by one-way ANOVA followed by the Tukey-Kramer *post hoc* test) (**Fig 3D, S5B Fig of S1 Data**).

These results indicate that BO-induced cardiac fibrosis and myocyte apoptosis might be mediated, at least in part, through the activation of β-adrenergic receptor (β-AR) signaling. Importantly, BO-induced cardiac fibrosis and myocyte apoptosis were blocked by co-treatment with propranolol.

## Bax expression was increased and Bcl-2 expression was decreased in the heart of BO mice

Expression of Bax, an accelerator of apoptosis, in the heart was significantly increased by BO treatment (Control ($n$ = 4) vs. BO ($n$ = 5); 100 ± 8.1 vs. 191 ± 51%, $P$ = 1.6 x $10^{-2}$ vs. Control by one-way ANOVA followed by the Tukey-Kramer *post hoc* test) in accordance with the previous study (**S6A Fig of S1 Data**) [43]. Propranolol alone had no effect on Bax expression, but blocked the BO-induced increase (BO ($n$ = 5) vs. BO + Pro ($n$ = 6); 191 ± 51 vs. 118 ± 49%, $P$ = 3.6 x $10^{-2}$ vs. BO by one-way ANOVA followed by the Tukey-Kramer *post hoc* test) (**S6A Fig of S1 Data**).

We also found that the expression of Bcl-2, a decelerator of apoptosis, in cardiac muscle was significantly decreased by BO treatment (Control ($n$ = 4) vs. BO ($n$ = 4); 100 ± 19 vs. 55 ± 22%, $P$ = 3.5 x $10^{-2}$ by one-way ANOVA followed by the Tukey-Kramer *post hoc* test) in accordance with the previous study (**S6B Fig of S1 Data**) [43]. Propranolol alone had no effect on the Bcl-2 expression, but blocked the BO-induced decrease (BO ($n$ = 4) vs. BO + Pro ($n$ = 6); 55 ± 22 vs. 94 ± 17%, $P$ = 5.0 x $10^{-2}$ by one-way ANOVA followed by the Tukey-Kramer *post hoc* test) (**S6B Fig of S1 Data**).

## Effects of BO on necroptosis

Programmed necrosis, often referred as necroptosis, occurs in various cardiovascular diseases [44], and receptor-interacting protein 3 (RIP3) is a key determinant of necroptosis, in addition to apoptosis and inflammation, in various types of cells, including cardiac myocytes [33]. RIP3 expression in the heart was significantly increased in the BO group (Control ($n$ = 6) vs. BO ($n$ = 6); 100 ± 1.9 vs. 276 ± 126%, $P$ = 2.4 x $10^{-3}$ by one-way ANOVA followed by the Tukey-Kramer *post hoc* test), and propranolol blocked this increase (BO ($n$ = 6) vs. BO + Pro ($n$ = 6); 276 ± 126 vs. 142 ± 44%, $P$ = 2.2 x $10^{-2}$ by one-way ANOVA followed by the Tukey-Kramer *post hoc* test) (**Fig 4A, S7A Fig of S1 Data**).

Thus, necroptosis might contribute to the development of cardiac dysfunction following BO treatment through activation of the β-AR signaling pathway.

## Effects of BO on CaMKII phosphorylation

CaMKII was recently found to be one of the targets of receptor interacting protein 3 kinase (RIP3), which activates CaMKII via phosphorylation and oxidation [45]. Notably, sustained activation of CaMKII is recognized to promote heart failure [46,47]. We thus examined the amounts of phospho-CaMKII (Thr-286) in the heart of BO mice and found that it was significantly increased (Control ($n$ = 6) vs. BO ($n$ = 6); 100 ± 16 vs. 311 ± 83%, $P$ = 8.9 x $10^{-5}$ by one-way ANOVA followed by the Tukey-Kramer *post hoc* test) in accordance with the previous study (**S8 Fig of S1 Data**) [43]. Propranolol alone had no effect on the amounts of phospho-CaMKII (Thr-286), but propranolol blocked this increase (BO ($n$ = 6) vs. BO + Pro ($n$ = 6); 311 ± 83 vs. 127 ± 37%, $P$ = 4.6 x $10^{-4}$ by one-way ANOVA followed by the Tukey-Kramer *post hoc* test) (**S8 Fig of S1 Data**).

These data suggest that BO-induced cardiac dysfunction might be mediated, at least in part, via RIP3/CaMKII signaling downstream of β-AR activation.

## Effects of BO on PLN phosphorylation

The importance of PLN regulation of sarcoendoplasmic reticulum calcium transport ATPase (SERCA) function for cardiac muscle health and in disease is well established [21]. We thus examined the effects of BO on PLN phosphorylation in cardiac muscle, focusing on Thr-17, which is phosphorylated by CaMKII, and Ser-16, which is phosphorylated by protein kinase A (PKA). Phospho-PLN (Thr-17) and phospho-PLN (Ser-16) were significantly increased in cardiac muscle of BO mice (PLN (Thr-17): Control ($n$ = 5) vs. BO ($n$ = 4): 100 ± 16 vs. 165 ± 36%, $P$ = 3.2 x $10^{-2}$ by one-way ANOVA followed by the Tukey-Kramer *post hoc* test; PLN (Ser-16): Control ($n$ = 5) vs. BO ($n$ = 5): 100 ± 35 vs. 205 ± 57%, $P$ = 2.1 x $10^{-2}$ by one-way ANOVA followed by the Tukey-Kramer *post hoc* test) in accordance with the previous study (**S9A and S9B Fig of S1 Data**) [43]. Propranolol alone had no effect on the amounts of phospho-PLN (Thr-17 and Ser-16), but propranolol blocked both phosphorylations (PLN (Thr-17): BO ($n$ = 4) vs. BO + Pro ($n$ = 5); 165 ± 36 vs. 57 ± 40%, $P$ = 5.6 x $10^{-4}$ by one-way ANOVA followed by the Tukey-Kramer *post hoc* test; PLN (Ser-16): BO ($n$ = 5) vs. BO + Pro ($n$ = 6); 205 ± 57 vs. 97 ± 45%, $P$ = 1,2 x $10^{-2}$ by one-way ANOVA followed by the Tukey-Kramer *post hoc* test) (**S9A and S9B Fig of S1 Data**).

These data suggest that BO-induced cardiac fibrosis and apoptosis might be induced, at least in part, through β-AR-mediated activation of PLN phosphorylation on threonine 17, as well as on serine 16.

## Effects of BO on Akt/mTORC1 phosphorylation

We then examined the effects of BO on Akt/mTORC1 signaling (**Fig 4B and 4C**), which is known to be cardioprotective in multiple cardiac pathological conditions [48–51]. Akt phosphorylation (Ser-473) of cardiac muscle was significantly decreased in the heart of BO mice (Control ($n$ = 6) vs. BO (n = 4): 100 ± 15 vs. 70 ± 13%, $P$ = 1.0 x $10^{-2}$ by the Tukey-Kramer *post hoc* test). Propranolol blocked this decrease (BO ($n$ = 4) vs. BO + Pro ($n$ = 6); 70 ± 13 vs. 103 ± 10%, $P$ = 5.1 x $10^{-3}$ by one-way ANOVA followed by the Tukey-Kramer *post hoc* test) (**Fig 4B, S7B Fig of S1 Data**).

mTOR phosphorylation on serine 2448, a specific marker of mTORC1 formation, was also significantly decreased in the heart of BO mice (Control ($n$ = 6) vs. BO (n = 6): 100 ± 8 vs. 64 ± 16%, $P$ = 9.8 x $10^{-5}$ by one-way ANOVA followed by the Tukey-Kramer *post hoc* test). Again, propranolol blocked this decrease (BO ($n$ = 6) vs. BO + Pro ($n$ = 6); 64 ± 16 vs. 93 ± 6%, $P$ = 3.8 x $10^{-2}$ by one-way ANOVA followed by the Tukey-Kramer *post hoc* test) (**Fig 4C, S7C Fig of S1 Data**).

These data suggest that BO-mediated cardiac dysfunction might be mediated, at least in part, through the inhibition of Akt/mTORC1 signaling.

## Effects of BO on mTORC2 phosphorylation

We also found that mTOR phosphorylation at Ser-2481, a specific marker of mTORC2 formation for the cyclic AMP (cAMP)/PKA signaling pathway in skeletal muscle [26,52], was also significantly increased in the heart of BO mice (Control ($n$ = 6) vs. BO (n = 6): 100 ± 14 vs. 326 ± 291%, $P$ = 2.6 x $10^{-2}$ by one-way ANOVA followed by the Tukey-Kramer *post hoc* test) (**Fig 5A, S10A Fig of S1 Data**). Propranolol blocked this increase (BO ($n$ = 6) vs. BO + Pro ($n$ = 6); 326 ± 291 vs. 102 ± 44%, $P$ = 2.8 x $10^{-2}$ by one-way ANOVA followed by the Tukey-Kramer *post hoc* test) (**Fig 5A, S10A Fig of S1 Data**).

These results suggest that the increase of mTORC2 phosphorylation might also be involved in BO-induced cardiac dysfunction.

## Effects of BO on autophagic activity

We next investigated the effects of BO on autophagy in the heart, because the basal level of autophagy is important to maintain physiological muscle homeostasis, and autophagy also plays a role in the response to stress [53].

The amount of microtubule-associated protein light chain 3-II (LC3-II), which is correlated with the number of autophagosomes [54], was significantly decreased in the heart of BO mice (Control ($n = 6$) vs. BO ($n = 6$): 100 ± 3 vs. 53 ± 11%, $P = 1.8$ x $10^{-3}$ by one-way ANOVA followed by the Tukey-Kramer *post hoc* test). Propranolol blocked this decrease (BO ($n = 6$) vs. BO + Pro ($n = 6$): 53 ± 11 vs. 93 ± 14%, $P = 1.0$ x $10^{-2}$ by one-way ANOVA followed by the Tukey-Kramer *post hoc* test) (**Fig 5B**, **S10B Fig of S1 Data**).

Besides LC3, total cellular expression levels of p62 is inversely correlated with the autophagic degradative activity, i.e., autophagic flux. We found that p62 expression was significantly increased in the BO group (Control ($n = 6$) vs. BO ($n = 6$): 100 ± 2 vs. 207 ± 31%, $P = 3.7$ x $10^{-6}$ by one-way ANOVA followed by the Tukey-Kramer *post hoc* test), but this increase was blocked by propranolol (BO ($n = 6$) vs. BO + Pro ($n = 6$): 207 ± 31 vs. 126 ± 34%, $P = 1.2$ x $10^{-4}$ by one-way ANOVA followed by the Tukey-Kramer *post hoc* test) (**Fig 5C**, **S10C Fig of S1 Data**).

These data suggest that BO decreases not only the number of autophagosomes, but also the autopahgic flux in the heart via activation of β-AR.

## Effects of BO on oxidative stress

RIP3-induced CaMKII phosphorylation triggers opening of the mitochondrial permeability transition pore and myocardial necroptosis, in addition to apoptosis and inflammation, leading to oxidative stress-induced myocardial damage and heart failure [54]. We thus evaluated oxidative stress in the myocardium by means of 8-OHdG immunostaining (**Fig 6A and 6B**) and western blotting of oxidized proteins (**Fig 6C and 6D**). In order to confirm the validity of the immunostaining for 8-OHdG, we first prepared positive and negative control sections by incubating with (positive control)/without (negative control) 0.3% $H_2O_2$ in TBS-T for 1 h at room temperature before the anti-8-OHdG antibody treatment and confirmed that the 8-OHdG staining procedure used in this study could clearly discriminate 8-OHdG-positive and non-positive nuclei (**S11C Fig of S1 Data**).

The ratio of 8-OHdG-positive/total cells was significantly increased in the BO group (Control ($n = 6$) vs. BO ($n = 6$): 1.9 ± 1.1 vs. 14.1 ± 2.9%, $P = 3.5$ x $10^{-11}$ by one- way ANOVA followed by the Tukey-Kramer *post hoc* test), and the increase was blocked by propranolol (BO ($n = 6$) vs. BO + Pro ($n = 6$): 15.0 ± 2.9 vs. 2.1 ± 0.8%, $P = 5.2$ x $10^{-11}$ by one-way ANOVA followed by the Tukey-Kramer *post hoc* test) (**Fig 6A and 6B**, **S11A Fig of S1 Data**).

The amount of oxidized proteins, measured using the OxiSelect TM protein kit, was also significantly increased (Control ($n = 6$) vs. BO ($n = 6$): 100 ± 6.2 vs. 383 ± 245%, $P = 3.8$ x $10^{-3}$ by one-way ANOVA followed by the Tukey-Kramer *post hoc* test), and again the increase was blocked by propranolol (BO ($n = 6$) vs. BO + Pro ($n = 6$): 383 ± 245 vs. 96 ± 25%, $P = 3.4$ x $10^{-3}$ by one-way ANOVA followed by the Tukey-Kramer *post hoc* test) (**Fig 6C and 6D**, **S11B Fig of S1 Data**).

These results indicate that BO treatment increases oxidative stress-induced myocardial damage, which might contribute to the cardiac dysfunction in BO mice.

## Discussion

Our aim was to evaluate potential effects of occlusal disharmony on cardiac homeostasis using BO mice, which have been used in research on occlusal disharmony previously by us and

other groups [8,9,15]. Our research was motivated by existing knowledge of the impact of psychological and physical stress, and subsequent increase of sympathetic nerve activity, on the development of cardiovascular disease in humans, even though the mechanism remains poorly understood [55,56]. We first confirmed that BO increases stress by measuring corticosterone levels and altered HR control by the autonomic nervous system in BO mice. Secondary effects on the heart include increases of cardiac fibrosis, cardiac myocyte apoptosis and oxidative stress with decreased cardiac function and altered signal transduction in cardiac muscle.

Although acute sympathetic stimulation is a major mechanism to improve cardiac dysfunction, chronic sympathetic stimulation, as typically seen in heart failure, induces cardiac myocyte apoptosis, which leads to further deterioration of cardiac function and intensification of heart failure [21,28,30,51]. Recently, it has been shown that chronic sympathetic stimulation activates not only the cAMP/PKA pathway, but also cAMP/exchange protein directly activated by cAMP (Epac) pathway [57,58]. More recently, we showed that increased PLN phosphorylation on serine 16, a major target of cAMP/PKA and the cAMP/Epac pathway, and on threonine 17, a major target of the CaMKII pathway leading to enhanced $Ca^{2+}$ leakage from sarcoplastic reticulumn, may cause cardiac dysfunction in responses to various stresses [21]. Our current findings, together with the previous studies, indicated that BO-induced cardiac dysfunction might be mediated through the activation of β-AR. Also, BO-induced alteration of cardiac homeostasis was completely blocked by co-treatment with the non-selective β-AR blocker propranolol [59].

We also carried out HRV analysis and compared the ratio of LF/HF, an index of the sympathetic nervous activity [25], nHF, an index of the parasympathetic nervous activity, and SDNN, which is a measure of total autonomic instability [24,25,60], at 1 day before (baseline) and at 1, 7 and 14 days after BO treatment. The LF/HF ratio was significantly increased and nHF was significantly decreased, compared to the baseline at all time points after BO, as expected, because occlusal disharmony increases stress in humans [61,62] and in rats [6]. However, mean HR was unexpectedly reduced in BO mice, compared to the control baseline. Recently, it has been demonstrated that rats exposed to stress exhibit significantly increased serum corticosterone levels (> 200 ng/mL from baseline) to the same degree as BO mice, and show decreased HR, compared to the baseline, even if the LF/HF ratio is increased and nHF is decreased, as observed in BO mice [63]. SDNN reflects the balance between the sympathetic and parasympathetic inputs to the cardiac pacemaker and thus SDNN is also a measure of total autonomic instability [24,25,60]. We do not completely understand the mechanisms that contribute to the decreased HR after BO treatment. We have previously demonstrated that SDNN and R-R interval are significantly increased under microgravity stress in type 5 adenylyl cyclase (a major cardiac and Gi-inhibitable isoform) null mice, which show loss of parasympathetic restraint [24,30]. Since behavioral and physiological flexibility to respond to stress depend upon parasympathetic modulation, our results indicated that decreased HR after the BO treatment might be mediated through the altered parasympathetic modulation in the heart of BO mice.

Previous clinical and experimental findings support a major role for activation of the sympathetic nervous system and parasympathetic nervous withdrawal in the genesis of heart failure and in heart failure progression [21,28,51,64,65]. This autonomic imbalance exerts adverse effects on the heart, blood vessels, and kidneys, resulting in pathological left ventricular remodeling, peripheral vasoconstriction, and salt and water retention, respectively [66,67]. These observations, along with the success of β-AR blockade in the treatment of heart failure, provide a rationale for therapies that inhibit adrenergic activity, enhance parasympathetic activity, or, preferably, accomplish both, leading to a decreased risk of death as well as hospitalization for cardiovascular causes in patients with heart failure [59,67,68]. This study, together

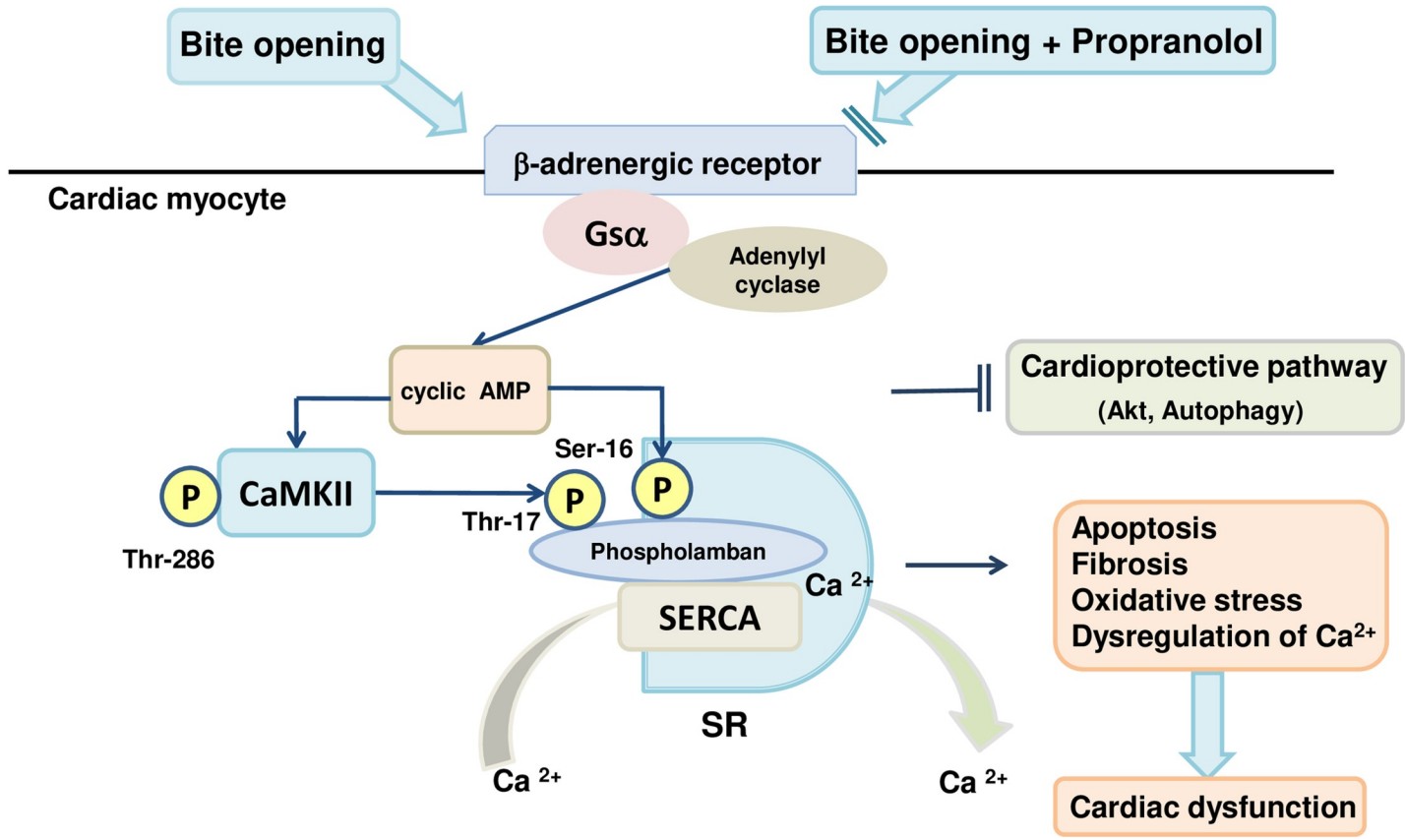

**Fig 7. Schematic illustration of the proposed role of β-AR signaling in cardiac muscle.** This scheme illustrates the proposed role of β-AR signaling in the heart of BO-treated mice. β-AR signaling is activated by the BO treatment, leading to the activation of CaMKII-mediated PLN phosphorylation (Thr-17), and cAMP/PKA -mediated PLN phosphorylation (Ser-16), but cardioprotective signaling such as Akt and autophagic signaling was decreased by the BO treatment (*left*). On the other hand, co-treatment with propranolol (*right*) protected the heart from BO-induced cardiac dysfunction. CaMKII: calmodulin kinase II, SERCA2a: sarcoendoplasmic reticulumn (SR) calcium transport ATPase.

with the previous studies, indicated that occlusal disharmony might cause cardiovascular disease through the disturbances in sympathetic/parasympathetic neural regulation, and β-AR blockade might reduce the risk of occlusal disharmony-mediated cardiovascular diseases.

In conclusion, our results indicate that occlusal disharmony-induced stress leads to cardiac dysfunction through the activation of sympathetic nerve activity, which can be blocked by propranolol. The cardiac dysfunction is mediated by increased PLN phosphorylation at threonine 17 and serine 16, as well as inhibition of Akt/mTOR signaling and autophagic activity, leading to cardiac fibrosis, cardiac myocyte apoptosis, and oxidative stress (**Fig 7**). Consequently, occlusal disharmony might alter cardiac homeostasis through activation of the sympathetic nervous system and reduction of the parasympathetic influence on the heart.

## Supporting information

**S1 Data.**
(PDF)

**S2 Data.**
(PDF)

## Author Contributions

**Conceptualization:** Yuka Yagisawa, Kenji Suita, Yoshiki Ohnuki, Satoshi Okumura.

**Formal analysis:** Yuka Yagisawa, Kenji Suita, Yasuharu Amitani, Satoshi Okumura.

**Funding acquisition:** Satoshi Okumura.

**Investigation:** Yuka Yagisawa, Kenji Suita, Yoshiki Ohnuki, Yasumasa Mototani, Aiko Ito, Ichiro Matsuo, Yoshio Hayakawa, Megumi Nariyama.

**Methodology:** Yuka Yagisawa, Kenji Suita, Yoshiki Ohnuki, Misao Ishikawa, Daisuke Umeki.

**Supervision:** Yasutake Saeki, Yoshiki Nakamura, Hiroshi Tomonari.

**Writing – original draft:** Yuka Yagisawa, Satoshi Okumura.

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
