## [Decision Letter · Decision Letter 0]

2 Dec 2019

PONE-D-19-25337

Effects of occlusal disharmony on cardiac function in mice

PLOS ONE

Dear Dr. Okumura,

Thank you for submitting your manuscript to PLOS ONE. After careful consideration, we feel that it has merit but does not fully meet PLOS ONE’s publication criteria as it currently stands. Therefore, we invite you to submit a revised version of the manuscript that addresses the points raised during the review process.

The manuscript has been assessed by two reviewers, their comments are available below.

The reviewers have raised a number of major concerns about the study that need attention in a revision. The reviewers note that the reporting of the methodology should be improved, they request that you adhere to the reporting requirements of the ARRIVE guidelines and they note the manuscript should report methods used for anesthesia, analgesia and euthanasia. The reviewers note concerns about claims around chronic stress as the manuscript has not reported data on stress markers and heart rate variability during two weeks for the different groups. The reviewers also note that revisions are needed on the statistical analyses.

Could you please carefully revise the manuscript to address the items raised.

We would appreciate receiving your revised manuscript by Jan 13 2020 11:59PM. To enhance the reproducibility of your results, we recommend that if applicable you deposit your laboratory protocols in protocols.io, where a protocol can be assigned its own identifier (DOI) such that it can be cited independently in the future. For instructions see: http://journals.plos.org/plosone/s/submission-guidelines#loc-laboratory-protocols

A rebuttal letter that responds to each point raised by the academic editor and reviewer(s). This letter should be uploaded as separate file and labeled 'Response to Reviewers'.A marked-up copy of your manuscript that highlights changes made to the original version. This file should be uploaded as separate file and labeled 'Revised Manuscript with Track Changes'.An unmarked version of your revised paper without tracked changes. This file should be uploaded as separate file and labeled 'Manuscript'

We look forward to receiving your revised manuscript.

Kind regards,

Iratxe Puebla

Senior Managing Editor, PLOS ONE

Journal Requirements:

2. To comply with PLOS ONE submissions requirements, in your Methods section, please provide additional information on the animal research and ensure you have included details on (1) methods of sacrifice, (2) methods of anesthesia and/or analgesia, and (3) efforts to alleviate suffering.

'The funders had no role in study design, data collection and analysis, decision to

publish, or preparation of the manuscript.'

Please provide an amended Funding Statement that declares *all* the funding or sources of support received during this specific study (whether external or internal to your organization) as detailed online in our guide for authors at http://journals.plos.org/plosone/s/submit-now.  

Please state what role the funders took in the study.  If any authors received a salary from any of your funders, please state which authors and which funder. If the funders had no role, please state: "The funders had no role in study design, data collection and analysis, decision to publish, or preparation of the manuscript."

Reviewers' comments:

Reviewer's Responses to Questions

**Comments to the Author**

1. Is the manuscript technically sound, and do the data support the conclusions?

Reviewer #1: No

Reviewer #2: No

2. Has the statistical analysis been performed appropriately and rigorously? 

Reviewer #1: Yes

Reviewer #2: No

3. Have the authors made all data underlying the findings in their manuscript fully available?

Reviewer #1: Yes

Reviewer #2: No

4. Is the manuscript presented in an intelligible fashion and written in standard English?

Reviewer #1: No

Reviewer #2: Yes

5. Review Comments to the Author

Reviewer #1: In this manuscript, the authors examined the effects of occlusal disharmony on cardiac function in bite-opening mice and tested the effects of propranolol. Differences shown in the results are clear, however, this reviewer has some concerns about description of the manuscript and design of the experiments. More specifically, 24 hours of heart rate monitoring just after 1 day from the treatment does not support that viewpoint that BO induces chronic sympathetic nerve activation even though beta blocker suppresses the all changes induced by the BO treatment.

Running title

Do the author’s findings really indicate heart failure in mice?

Method

Page 7 and following. Please indicate number of animals that were used in this study.

Page 8, Line 2. It would be better to include approval number from the internal committee in text, if applicable.

Page 8, Line 4. What timing did the authors perform physiological experiments?

Page 8, Line 5. How many animals and groups were tested?

Page 8, Line 5. Please indicate given dose of isoflurane, e.g. 2% isoflurane in room air, etc.

Page 8, Line 10. How many animals and groups were tested? Please clarify.

Page 8, Line 10. What type of transmitter was used? Did the authors really implant it into “consciousness” mice without any of anesthesia and analgesia?

Page 8, Line 16. Why did the authors decide to record ECG data at 1 day after the BO treatment? Why not 7 days later?

Page 8, Line 17. “HRV” should be specified.

Page 9, Line 6. Please check the sentence.

Page 9, Line 10. What was the method for euthanasia?

Page 13, Line 2. “8-OH-dG”

Page 13, Line 6. If applicable, please indicate what application was used to perform the statistical analysis.

Results

Overall, this section is too wordy, and some sentences can be moved to discussion section.

Page 14, Line 15. Mice were group housed. Was food and water consumption measured as average of 3-4 mice? If so, please indicate in the text.

Page 15, Line 11. Please show time course changes in heart rate during 24 h of ECG measurement (circadian rhythm). Did heart rate increase in BO mice? and did it last for further 14 days? Do the authors think that propranolol treatment affects these changes?

Page 16, Line 7. CA to indicate “cardiac muscle mass” does not make sense.

Page 16, Line 13. Please reconsider this sentence.

Page 16, Line 16. Please explain or discuss why did these changes occur. Do the authors know or have an evidence that acute increase in sympathetic activity decrease ejection fraction in the heart?

Page 16, Line 17. “Beta-AR” should be specified.

Page 18, Line 1. This sentence might be too speculative since the authors used non-selective beta blocker, propranolol.

Discussion

The results were not well discussed in this section. Please discuss what does your data mean here.

Figure1

In figure 1D and 1E, error bars are missing.

After BO, LH/HF decreased toward the baseline level within 48 h from BO treatment. This reviewer guesses this may not be chronic increase in sympathetic nerve activity. Do the authors think that LF/HF returns to normal level or maintained at high level at 7 days and 14 days from the BO treatment?

Reviewer #2: The paper shows that propranolol changed cardiac fibrosis in a occlusal disharmony model.

This is an interesting study. However, I would like to make some points regarding the manuscript. The article needs to be revised. First, there were no data of any stress markers and heart rate variability (HRV) during 2 weeks in four groups. Second, the paper should be followed by the ARRIVE Guidelines and use the checklist.

TITLE

1. The present study did not investigate cardiac function but only investigate fibrosis and protein expression at one time. Please revise the title.

ABSTRACT

1. What is the main outcome in this study? Because there were no data of any stress markers and heart rate variability during 2 weeks in four groups, the authors should re-consider the main outcome and then, revise the abstract and text.

2. The conclusion is not appropriate because this study did not investigate the orthodontic treatment.

INTRODUCTION

1. The authors did not investigate the cardiac function as above.

2. The experimental period was only 2 weeks. It does not fit to “chronic stress”. Furthermore, the authors did not investigate the stress in this study.

MATERIALS AND METHODS

1. The authors should add more detail parts according to the ARRIVE Guidelines and checklist (see above).

2. There were no comments about killing methods and fixation. This reviewer thought that the authors had used Bouin's Fluid because they detected 8-OHdG expression. Please add the details in the text.

3. Please add the approval number in the text (P8).

4. Did the authors collect the blood sample? If yes, please investigate the status of stress using blood samples.

5. Did the authors perform the sample size estimation?

6. Please add the results of any stress markers and heart rate variability (HRV) during 2 weeks in four groups (see above)

7. In western blotting, the authors should clarify the dilution and reference of each antibody (P11).

8. What do the authors mean “after paraffinization with 4% (v/v) paraformaldehyde”? (P12) They should revise the sentence carefully. Second, fixation by 4% (v/v) paraformaldehyde affect the staining for 8-OHdG and it is not recommended. Why did they use 4% (v/v) paraformaldehyde? Third, they have to perform antigen retrieval following the guideline when they use 4% (v/v) paraformaldehyde.

9. Please add some comments about validity, reliability and reproducibility in each procedure.

10. Were the all data parametric? Did the authors check it? When the number was three or four, it was too small in t-test.

11. In the time course analyses (Figure 1 and Supplement), the authors should use two-way ANOVA or other analyses but not t-test or one-way ANOVA.

RESULTS

1. The results will be changed by new methods.

2. Please show the original p value but not “P<0.05” or NS.

3. The authors should add the data; i.e., BO vs. Pro, BO vs. BO+Pro, and Pro vs. BO+Pro in all figures because they use the Tuckey-Kramer test.

4. The number of each group in the Table 1 was wrong. Second, please add the full names, BW, BO, and Pro. Furthermore, please add the statistical name. Please revise them.

5. The Figure 2C was unclear. Please change it.

6. Why was the number of each group different among figures? Please clarify it in the text.

DISCUSSION

1. Please revise the Figure 7 to avoid the misleading. The model does not reflect a chronic stress model and the authors did not investigate stress markers and HRV during the 2 weeks among all groups.

2. Please delete the comments about periodontal disease (P25) and orthodontic treatment (P26) because the authors did not investigate the effects.

3. Please revise the discussion following the new results or the guideline.

6. PLOS authors have the option to publish the peer review history of their article (what does this mean?). If published, this will include your full peer review and any attached files.

Reviewer #1: No

Reviewer #2: No

---

## [Author Response · Author response to Decision Letter 0]

4 Jun 2020

Reviewer#1:

In this manuscript, the authors examined the effects of occlusal disharmony on cardiac function in bite-opening mice and tested the effects of propranolol. Differences shown in the results are clear, however, this reviewer has some concerns about description of the manuscript and design of the experiments. More specifically, 24 hours of heart rate monitoring just after 1 day from the treatment does not support that viewpoint that BO induces chronic sympathetic nerve activation even though beta blocker suppresses the all changes induced by the BO treatment.

1. Running title

1-(1). Do the author’s findings really indicate heart failure in mice?

Responses: We modified the running title as follows.

Intraoral mechanical stress and β-adrenergic signaling

2. Method

2-(1). Page 7 and following. Please indicate number of animals that were used in this study. 

Responses: We incorporated the number of groups and animals in each experiment into the methods section of the revised manuscript (Page 7, Line 1-Page 16, Line 8).

2-(2). Page 8, Line 2. It would be better to include approval number from the internal committee in text, if applicable.

Responses: We incorporated the required information in the revised manuscript as shown below (Page 8, Lines 3-4).

The experimental protocol was approved by the Animal Care and Use Committee of Tsurumi University (No. 29A041)

2-(3). Page 8, Line 4. What timing did the authors perform physiological experiments?

Responses: We incorporated the following sentences in the revised manuscript (Page 8, Line 17-Page 9, Line 2).

--- echocardiographic measurements (Control: n = 10, BO: n = 7, Pro: n = 5; BO + Pro: n = 7) were performed by means of ultrasonography (TUS-A300, Toshiba, Tokyo, Japan) at 14 days after the BO treatment [1].

2-(4). Page 8, Line 5, How many animals and groups were tested?

Responses: We incorporated the number of groups in the revised manuscript as shown below (Page 7, Lines 9-11). 

---were divided into four groups: a normal control group (Control), a BO-only treatment group (BO), a propranolol-only treatment group (Pro) and a BO plus propranolol treatment group (BO + Pro) (Fig 1A).---

We also incorporated the number of animals in each experiment into the methods section, in response to comment 2-(1) from this reviewer (Page 7, Line 1-Page 16, Line 8).

2-(5). Page 8, Line 5. Please indicate given dose of isoflurane, e.g. 2% isoflurane in room air, etc.

Responses: We incorporated the required information in the revised manuscript as shown below (Page 8, Lines 16-17).

Mice were anesthetized via a mask with isoflurane (1.0-1.5% v/v) at room temperature --- 

2-(6). Page 8, Line 10. How many animals and groups were tested? Please clarify.

Responses: We incorporated the required information in the revised manuscript as shown below (Page 9, Line 17- Page 10, Line 1).

---a transmitter (F20-EET; Data Sciences International, St. Paul, MN, USA) was implanted into the mice (n = 5) at 14 days before the BO treatment.

2-(7)-1). Page 8, Line 10. What types of transmitter was used? 

Responses: Please see the response to comment 2-(6) from this reviewer.

2-(7)-2). Did the authors really implant it into “consciousness” mice without any of anesthesia and analgesia?

Responses: We apologize for the error. We incorporated the following sentences in the revised manuscript (Page 9, Lines 15-Page, Line 1).

Mice were anesthetized with intraperitoneal medetomidine (0.03 mg/ml), midazolam (0.4 mg/ml), and butorphanol (0.5 mg/ml). Then, an abdominal midline incision was made on the ventral surface, and a transmitter (F20-EET; Data Sciences International, St. Paul, MN, USA) was implanted into the mice (n = 5) at 14 days before the BO treatment. 

2-(8). Page 8, Line 16. Why did the authors decide to record ECG data at 1 day after the BO treatment? Why not 7 days later?

Responses: We recorded ECG data for 24 h at 1 day before the BO treatment to obtain the baseline, and then at 1, 7 and 14 days after the BO treatment in the revised manuscript. 

We incorporated the following sentences in the revised manuscript as shown below (Page 10, Lines 5-6). 

ECG data were recorded for 24 h at 1 day before the BO treatment to obtain the baseline and at 1, 7 and 14 days after the BO treatment (Fig 2A) [2,3].

We also incorporated the above data into the results section (Page 18, Line 12-Page 20, Line 17) and Fig 2C-G of the revised manuscript.

2-(9). Page 8, Line 17. “HRV” should be specified.

Responses: “HRV” is an abbreviation for heart rate variability. We defined this abbreviation in the revised manuscript (Page 10, Line 7-8).

2-(10). Page 9, Line 6. Please check the sentence.

Responses: Thank you. We modified the sentence as shown below in the revised manuscript (Page 10, Lines 14-15).

---as a marker of parasympathetic activity to examine the effects of BO treatment

[2,3].

2-(11). What was the method for euthanasia?

Response: We incorporated the following sentences in the revised manuscript with new references (Page 9, Lines 2-4).

After the completion of echocardiographic measurement, mice were anesthetized via a mask with isoflurane (1.0-1.5% v/v) at room temperature and killed by cervical dislocation [4,5].

2-(12). Page 13, Line 2. “8-OH-dG”

Response: “8-OHdG” is an abbreviation for 8-hydroxy-2’-deoxyguanosine. We specified this abbreviation in the revised manuscript (Page 14, Line 6).

2-(13). Page 13, Line 6. If applicable, please indicate what application was used to perform the statistical analysis.

Response: We incorporated the required information in the revised manuscript with a new reference as shown below (Page 16, Lines 4-8).

Analyses were performed with PASW statistics 18 (SPSS Inc., Chicago, IL, USA) except for the sample size estimation, which was performed by G*Power version 3.1. (program, concept and design by Franz, Universitat Kiel, Germany; freely available windows application software) [6].

3. Results

3-(1). Overall, this section is too wordy, and some sentences can be moved to discussion section. 

Response: We have made the suggested changes. 

3-(2). Page 14, Line 15. Mice were group housed. Was food and water consumption measured as average of 3-4 mice? If so, please indicate in the text.

Response: Yes. We incorporated the following sentences in the revised manuscript (Page 17, Lines 12-14).

---consumption of pellet food and water per mouse, measured as an average of group-housed mice in each cage (approximately 3), during the 2-week experimental period. 

3-(3). Page 15, Line 11. Please show time course changes in heart rate during 24 h of ECG measurement (circadian rhythm). Did heart rate increase in BO mice? And did it last for further 14 days? Do the authors think that propranolol treatment affects these changes?

Response: We incorporated the required data in the revised manuscript as shown below (Page 19, Line 8-Page 20, Line 7).

Effects of BO on HR

To evaluate changes of HR, we examined the mean (Fig 2E, Fig S3D) and circadian variation of HR (Fig 2G, Fig S4B) at 1 day before the BO treatment to obtain the baseline (BO-1day) and at 1 (BO+1 day), 7 (BO+7day) and 14 days (BO+14day) after the BO treatment. 

Mean HR was unexpectedly but significantly decreased at 1, 7 and 14 days after the treatment of BO, compared to the baseline (BO-1day vs. BO+1day, P = 3.2 x 10-2; BO-1day vs BO+7day, P = 3.6 x 10-5; BO-1day vs. BO+14day, P = 3.6 x 10-2 by one- way repeated-measures ANOVA followed by the Bonferroni post hoc test, n = 5 each) (Fig 2E, Fig S3D). 

We also examined the circadian variation of HR and found that it was also decreased by the BO treatment (BO-1day vs. BO+1day, P = 1.6 x 10-56; BO-1day vs BO+7day, P = 1.2 x 10-44; BO-1day vs. BO+14day, P = 7.1 x 10-19 by two-way repeated-measures ANOVA followed by the the Bonferroni post hoc test, n = 5 each) (Fig 2G, Fig S4B).

These data suggest that BO treatment alters the control of HR via the autonomic nervous system.

3-(4). Page 16, Line 7. CA to indicate “cardiac muscle mass” does not make sense.

Response: “Cardiac muscle mass” is abbreviated as CMM in the revised manuscript (Page 9, Lines 6).

3-(5). Page 16, Line 13. Please reconsider this sentence.

Response: We modified the sentences as shown below (Page 21, Line 11-14).

However, no significant changes of LVEF and %FS were observed in the BO + Pro group compared to the control. Also, no significant differences of left ventricular end-diastolic (LVEDD) and left ventricular end-systolic diameter (LVESD) were observed,---

3-(6). Page 16, Line 16. Please explain or discuss why did these changes occur. Do the authors know or have evidence that acute increase in sympathetic activity decrease ejection fraction in the heart?

Response: We modified these sentences and moved them to the discussion section (Page 32, Line 12-Page 33, Line 7).

Although acute sympathetic stimulation is a major mechanism to improve cardiac dysfunction, chronic sympathetic stimulation, as typically seen in heart failure, induces cardiac myocyte apoptosis, which leads to further deterioration of cardiac function and intensification of heart failure [1,7-9]. Recently, it has been shown that chronic sympathetic stimulation activates not only the cAMP/PKA pathway, but also cAMP/exchange protein directly activated by cAMP (Epac) pathway [10,11]. More recently, we showed that increased PLN phosphorylation on serine 16, a major target of the cAMP/PKA and the cAMP/Epac pathway, and on threonine 17, a major target of the CaMKII pathway leading to enhanced Ca2+ leakage from sarcoplastic reticulumn, may cause cardiac dysfunction in responses to various stresses [1]. Our current findings, together with the previous studies, indicated that BO-induced cardiac dysfunction might be mediated through the activation of β-AR. Also, BO-induced alteration of cardiac homeostasis was completely blocked by co-treatment with the non-selective β-AR blocker propranolol [12]. 

3-(7). Page 16, Line 17. “Beta-AR” should be specified.

Response: “Beta-AR” is an abbreviation for β-adrenergic receptor. We added this in the revised manuscript (Page 23, Lines 4-5).

3-(8). Page 18, Line 1. This sentence might be too speculative since the authors used non-selective beta blocker, propranolol.

Response: We modified the sentences as shown below (Page 23, Lines 3-6).

These results indicate that BO-induced cardiac fibrosis and myocyte apoptosis might be mediated, at least in part, through the activation of β-adrenergic receptor (β-AR) signaling. Importantly, BO-induced cardiac fibrosis and myocyte apoptosis were blocked by co-treatment with propranolol.

4. Discussion

4-(1). The results were not well discussed in this section. Please discuss what does your data mean here.

Response: We modified the discussion to make it clear, as suggested (Page 32, Line 1-Page 35, Line 14).

5. Figure

5-(1). In Figure 1D and 1E, error bar are missing.

Response: Regarding the HRV analysis, we increased the experimental number (n = 5) and followed the response for a 2-week period as shown in Fig 2C-G in the revised manuscript.

5-(2). After BO, LH/HF decreased toward the baseline level within 48 h from BO treatment. This reviewer guesses this may not be chronic increase in sympathetic nerve activity. Do the authors think that LF/HF returns to normal level or maintained at high level at 7 days and 14 days from the BO treatment?

Response: BO treatment increased the LF/HF by approximately 4-fold at 1 day after the BO treatment, and the LF/HF remained unchanged at 14 days.

We incorporated the following sentences in the revised manuscript as shown below (Page 18, Line 12-Page 20, Line 7).

Effects of BO on LF/HF and nHF

To evaluate changes in autonomic nervous activity, we carried out HRV analysis and compared the ratio of LF to HF (LF/HF), an index of the sympathetic nervous activity [3], at 1 day before (BO-1 day) and at 1 (BO+1 day), 7 (BO+7day) and 14 days (BO+14day) after BO treatment. LF/HF was significantly greater than baseline at all time points (P < 0.01 by one-way repeated-measures ANOVA followed by the Bonferroni post hoc test, n = 5 each) (Fig 2C, Fig S3B). HF power was normalized to account for differences in total time power (nHF), and nHF was examined as an index of parasympathetic activity [3]. After BO, nHF was significantly decreased from baseline at all time points (P < 0.01 by one-way repeated-measures ANOVA followed by the Bonferroni post hoc test, n = 5 each) (Fig 2D, Fig S3C). 

These data suggest that sympathetic nerve activity was increased but parasympathetic activity was decreased after BO treatment.

Reviewer #2:

This paper shows that propranolol changes cardiac fibrosis in an occlusal disharmony model. This is an interesting study. However, I would like to make some points regarding the manuscript. The article needs to be revised. First, there were no data of any stress markers and heart rate variability (HRV) during 2 weeks in four groups. Second, the paper should be followed by the ARRIVE GUIDELINES and use the checklist.

1. Title

1-(1). The present study did not investigate cardiac function but only investigate fibrosis and protein expression at one time. Please revise the title.

Response: We modified the title as follows.

Effects of occlusal disharmony on cardiac homeostasis in mice

2. Abstract

2-(1). What is the main outcome in this study? Because there were no data of any stress markers and heart rate variability during 2 weeks in four groups, the authors should re-consider the main outcome and then, revise the abstract and text.

Response: We appreciate the suggestion. In the revised manuscript, we examined serum corticosterone level, a key biomarker for stress, at baseline and 14 days after the BO treatment in the revised manuscript [13]. We also recorded ECG data for 24 h at 1 day before the BO treatment to obtain the baseline and at 1, 7 and 14 days after the BO treatment, and performed HRV analysis all time points in the revised manuscript in order to respond comments 3-(3) and 5-(2) of reviewer-1.

We have not incorporated HRV analysis of the four groups as this reviewer suggested, as we could not complete the work within the timeframe for PLoS One revision. However, we are currently working another project that includes the HRV analysis of BO mice with/without co-treatment with β-blocker, and we hope to publish this in due course.

We revised the abstract accordingly (Page 3, Line 1-Page 4, Line 4). In addition we revised the text by incorporating the following sentences in the methods and results of the revised manuscript, as the reviewer suggested.

1) Page 8, Lines 6-13 (methods)

Serum corticosterone measurements

The serum was separated from blood samples collected from the heart of the control (n = 5) and BO mice (n = 5) under anesthesia for 14 days after the BO treatment. Each blood sampling procedure was completed within 30 s from the time of contact with the mouse. The separated serum samples were frozen at -80ºC until measurement. The serum corticosterone levels were determined using a Corticosterone HS EIA kit (#AC-15F1; Immunodiagnostic Systems Ltd., Tyne & Wear, UK), according to the manufacturer’s instructions.

2) Page 18, Lines 5-10 (results)

Effects of BO on serum corticosterone levels

Comparison of the levels of serum corticosterone level, a key biomarker for stress [14,15], in the control and BO mice at 14 days after the BO treatment revealed a significantly increase of approximately 3.6-fold at 14 days after BO treatment (n = 5 each) (Fig 2B, Fig S3A). These data suggest that the mice are stressed at 14 days after the BO treatment.

2-(2). The conclusion is not appropriate because this study did not investigate the orthodontic treatment.

Response: We agree and have modified the conclusion in the abstract as shown below. 

Page 4, Lines 3-5 (abstract)

These data suggest that occlusal disharmony might affect cardiac homeostasis via alteration of the autonomic nervous system.

3. Introduction

3-(1). The authors did not investigate the cardiac function as above.

Response: We modified the sentences as follows (Page 6, Lines 10-13).

Therefore, the aim of this study was to examine the effects of occlusal disharmony on stress markers, heart rate (HR) control via the autonomic nervous system, systolic cardiac function, histology and signal transduction in the heart, using bite-opening (BO) mice, which have previously been used in research on occlusal disharmony [14,16,17].

3-(2). The experimental period was only two weeks. It does not fit to “chronic stress”. Furthermore, the authors did not investigate the stress in this study.

Response: We agree. We replaced the term “chronic stress” with “stress” in the revised manuscript. As mentioned above, we examined the effects of BO on serum corticosterone level and found that it was significantly increased by approximately 3.6-fold at 14 days after BO treatment (n = 5 each), confirming that the mice are stressed (Fig 2B).　

4. Materials and Method

4-(1). The authors should add more detail parts according to the ARRIVE GUIDELINEs and checklist (see above).

Response: We incorporated the following sentences in the revised manuscript (Page 7, Line 18-Page 8, Line 4).

All animal experiments complied with the ARRIVE guidelines [18] and were carried out in accordance with the National Institutes of Health guide for the care and use of laboratory animals [19] and institutional guidelines. The experimental protocol was approved by the Animal Care and Use Committee of Tsurumi University (No. 29A041).

4-(2). There were no comments about killing methods and fixation. This reviewer thought that the authors had used Bouin’s Fluid because they detected 8-OHdG expression. Please add the details in the text. 

Response: Regarding the killing method, please see the response to comment 2-(11) from reviewer-1. 

We did not use Bouin’s Fluid. For fixation, we used 4% paraformaldehyde as in the previous study [20]. In order to confirm the validity of the immunostaining for 8-OHdG, we prepared positive and negative control sections by incubating with (positive control)/without (negative control) 0.3% H2O2 in TBS-T for 1 h at room temperature after the antigen retrieval, as described below. We confirmed that the 8-OHdG staining procedure can clearly discriminate 8-OHdG-positive and non-positive nuclei (Fig S9C).

We described the method of 8-OHdG staining in the revised manuscript as follows.

1) Page 14, Line 4-Page 15, Line 5 (method)

Immunostaining

Oxidative DNA damage in the myocardium was evaluated by immunostaining for 8-hydroxy-2’-deoxyguanosine (8-OHdG) using the Vector M.O.M Immunodetection system (#PK-2200, Vector Laboratories, Inc. Burlingame, CA, USA) [20,21]. Cross sections (Control; n = 6, BO; n = 6, Pro; n = 6, BO + Pro; n = 6), were cut with a cryostat at -20ºC at 10 μm, air-dried and fixed with 4% paraformaldehyde (v/v) in TBS-T for 5 min at room temperature. Antigen retrieval was achieved with 0.1% citrate plus 1% Triton X-100 for 30 min at room temperature, then the sections were washed with TBS-T, incubated with 0.3% horse serum in TBS-T for 1 h at room temperature, and blocked with M.O.M. blocking reagents (Vector Laboratories, Burlingame, CA, USA) overnight at 4ºC. For the positive control, sections were incubated with 0.3% H2O2 in TBS-T before the anti-8-OHdG antibody treatment (8.3 μg/ml in M.O.M. Dilute; clone N45.1 monoclonal antibody; Japan Institute for the Control of Aging, Shizuoka, Japan) overnight at 4ºC in a humidified chamber, and then incubated with 0.3% H2O2 in 0.3% horse serum for 1 h at room temperature to inactivate endogenous peroxidase, rinsed with TBS-T, incubated with anti-mouse IgG in M.O.M. Diluent, and processed with an ABC kit (Vector Laboratories, Inc. Burlingame, CA, USA). We calculated the ratio of 8-OHdG nuclei with oxidative DNA damage (stained dark blown) per total cell numbers

2) Page 30, Lines 2-7 (results)

In order to confirm the validity of the immunostaining for 8-OHdG, we first prepared positive and negative control section by incubating with (positive control)/without (negative control) 0.3% H2O2 in TBS-T for 1 h at room temperature before the anti-8-OHdG antibody treatment and confirmed that the 8-OHdG staining procedure used in this study could clearly discriminate 8-OHdG-positive and non-positive nuclei (Fig S9C).

4-(3). Please add the approval number in the text (P8). 

Response: Done as requested. 

4-(4). Did the authors collect the blood sample? If yes, please investigate the status of stress using blood sample.

Response: We did not collect blood samples for the four groups. However, to respond to this reviewer, we prepared control (n = 5) and BO-treated mice (n = 5), harvested the blood samples at 2 weeks, and examined serum corticosterone level, as a biomarker for stress [13], as mentioned above. Please see the response to comment 2-(1) from reviewer-2.

4-(5). Did the authors perform the sample size estimation?

Response: We calculated the total sample size of animals required for an ɑ risk of 0.05 and a statistical power (1-β) of 0.8 [22] by means of G*Power version 3.1. (program, concept and design by Franz, Universitat Kiel, Germany; freely available Windows application software) [6]. We incorporated the total sample size of all data in the revised manuscript (see Supplementary figures). 

However, the total sample sizes required for the statistical analysis of cardiac muscle mass (CMM) (total sample size = 72), CMM/tibia (total sample size = 80), CMM/body weight (total sample size n=196), LVEDD (total sample size = 32) and LVESD (total sample size = 120) were insufficient (Fig S10-S11). We could not prepare enough mice to perform the additional experiments necessary to improve the statistical power within a reasonable timeframe. Instead, we have incorporated a comment on this issue as a study limitation in the revised manuscript as shown below.

1) Page 16, Lines 3-4 (method)

The total sample size of animals required for statistical validity was calculated for an ɑ risk of 0.05 and a statistical power (1-β) of 0.8 [22]. 

2) Page 21, Lines 2-6 (result)

We examined the effects of BO on heart size in terms of CMM (mg), CMM per tibial length ratio (mg/mm) and CMM per body weight ratio (mg/g) (Table 1, Fig S10), and they were similar in all four groups. However, we cannot rule out the possibility that the statistical power was insufficient to detect BO-mediated cardiac hypertrophy as the total sample sizes in these cases were insufficient to provide an ɑ risk of 0.05 and a statistical power (1-β) of 0.8 (Fig S10).

3) Page 21, Lines 12-17

---no significant differences of left ventricular end-diastolic (LVEDD) and left ventricular end-systolic diameter (LVESD) were observed, although we cannot rule out the possibility that the statistical power was insufficient to detect BO-mediated effects on LVEDD and LVESD due to the limited total sample sizes (Fig S11).

4-(6). Please add the results of any stress markers and heart rate variability (HRV) during 2 weeks in four groups (see above). 

Response: Please see the response to comment 2-(1) from reviewer-2 (stress markers) and comment 2-(8) from reviewer-1 (HRV).

4-(7). In western blotting, the authors should clarify the dilution and reference of each antibody (p11).

Response: We incorporated the required information in the revised manuscript (Page 12, Line 15-Page 13, Line 10) with new references.

4-(8)-1). What do the authors mean “after paraffinization with 4% (v/v) paraformaldehyde”? (p12). They should revise the sentence carefully. 

Response: We apologize. “Paraffinization” should have been “fixation”. 

Please see the response to the comment 4-(2) from reviewer-2.

4-(8)-2). Second, fixation by 4% (v/v) paraformaldehyde affect the staining for 8-OHdG and it is not recommended. Why did they use 4% (v/v) paraformaldehyde?

Response: Please see the responses to comment 4-(2) from reviewer-2.

4-(8)-3). Third, they have to perform antigen retrieval following the guideline when they use 4% (v/v) paraformaldehyde.

Response: Please see the responses to the criticism 4-(2) from the reviewer-2.

4-(9). Please add some comments about validity, reliability and reproducibility in each procedure.

Response: Thank you for this suggestion. The procedures used in this study were similar to those used in our previous work: echocardiography [1,23], HRV analysis [2,3], Masson-trichrome staining and TUNEL staining [1,24], western blotting [4,25] and immunostaining [4,24], and each method was validated for reliability and reproducibility for each procedure. 

We incorporated the following sentences in the method section of the revised manuscript as shown below (Page 15, Lines 7-11).

Method validation

The procedures used in this study were similar to those used in our previous work: echocardiography [1,23], HRV analysis [2,3], Masson-trichrome staining and TUNEL staining [1,24], western blotting [4,25] and immunostaining [4,24], and each method was validated for reliability and reproducibility for each procedure. 

4-(10). Were the all data parametric? Did the author check it? When the number was three or four, it was too small in t-test.

Response: Normality assumption was verified using the Shapiro-Wilk test for all data in the revised manuscript. We incorporated the following sentences in the method section of the revised manuscript (Page 16, Lines 1-2).

Normality assumption was verified using the Shapiro-Wilk test for all data.

5. Results.

5-(1). The results will be changed by new methods.

Response: 1) We recorded ECG data for 24 h at 1 day before the BO treatment to obtain the baseline and at 1, 7 and 14 days after the BO treatment in the revised manuscript. 

We carried out HRV analyses at all time points and incorporated them in the revised manuscript (Page 18, Line 12-Page 20, Line 17).

2) We examined serum corticosterone level, a biomarker for stress, at baseline and 14 days after the BO treatment in the revised manuscript (Page 8, Lines 6-13). 

3) We asked Dr. Amitani, a statistics expert and co-author in the revised manuscript, to support us in re-analyzing all of the data. The revised statistics analysis has been added to the revised manuscript (Page 15, Line 13-Page 16, Line 8 and supplementary data).

5-(2). Please show the original p value but not “P<0.05” or NS

Response: We incorporated the original p value in the results section of the revised manuscript. We also incorporated the original p value for all data in the supplementary data file. 

5-(3). The authors should add the data; i.e., BO vs. Pro, BO vs. BO+Pro, and Pro vs. BO+Pro in all figures because they use the tuckey-Kramer test.

Response: We incorporated the required information (BO vs. Pro, BO vs. BO + Pro, and Pro vs. BO + Pro) in all figures of the revised manuscript.

5-(4). The number of each group in the Table 1 was wrong.

Second, please add the full names, BW, BO, and Pro.

Furthermore, please add the statistical name. Please revise them.

Response: Thank you. We have carefully checked and corrected Table 1, as the reviewer suggested.

5-(5). The Figure 2C was unclear. Please change it.

Response: Thank you. We replaced it with a better one.

5-(6). Why was the number of each group different among figures? Please clarify it in the text.

Response: We incorporated the number of groups and animals used in each experiment into the methods section of the revised manuscript (Page 7, Line 1-Page 16, Line 8) in response to comment 2-(1) from reviewer-1. For western blotting, we prepared crude protein homogenate from cardiac muscle excised from six mice of each group. However, we excluded outlying mice with extremely low or high values, compared to others of the same groups, from the analysis. This is why the n number varies in western blotting figures (Fig 4-6).

We incorporated the following sentences in the method section of the revised manuscript.

1) Page 12, Lines 11-14

Equal amounts of protein (5 μg) (Control; n = 6, BO; n = 6, Pro; n = 6, BO + Pro; n = 6) were subjected to 12.5 % SDS-polyacrylamide gel electrophoresis and blotted onto 0.2 mm PVDF membrane (Millipore, Billerica, MA, USA). 

2) Page 13, Lines 17-Page 14, Line 2

The reason why there are different numbers of samples in different western blotting figures (Fig 4-6) is that we excluded outliers (extremely low or high values, compared to others in the same group).

6. Discussion

6-(1). Please revise the Figure 7 to avoid the misleading. The model does not a chronic stress model and the authors did not investigate stress markers and HRV during the 2 weeks among all groups.

Response: We revised Figure 8 (Figure 7 in the original version) in line with the reviewer’s comments. 

6-(2). Please delete the comments about periodontal disease (p25) and orthodontic treatment (p26) because the authors did not investigate the effects.

Response: We deleted these comments as the reviewer suggested. 

6-(3). Please revise the discussion following the new results or the guideline.

Response: We revised the discussion to take account of new results as the reviewer suggested.

References

1. Okumura S, Fujita T, Cai W, Jin M, Namekata I, et al. (2014) Epac1-dependent phospholamban phosphorylation mediates the cardiac response to stresses. J Clin Invest 124: 2785-2801.

2. Okumura S, Tsunematsu T, Bai Y, Jiao Q, Ono S, et al. (2008) Type 5 adenylyl cyclase plays a major role in stabilizing heart rate in response to microgravity induced by parabolic flight. J Appl Physiol (1985) 105: 173-179.

3. Bai Y, Tsunematsu T, Jiao Q, Ohnuki Y, Mototani Y, et al. (2012) Pharmacological stimulation of type 5 adenylyl cyclase stabilizes heart rate under both microgravity and hypergravity induced by parabolic flight. J Pharmacol Sci 119: 381-389.

4. Ohnuki Y, Umeki D, Mototani Y, Jin H, Cai W, et al. (2014) Role of cyclic AMP sensor epac1 in masseter muscle hypertrophy and myosin heavy chain transition induced by β2-adrenoceptor stimulation. J Physiol 592: 5461-5475.

5. Goodman CA, Frey JW, Mabrey DM, Jacobs BL, Lincoln HC, et al. (2011) The role of skeletal muscle mTOR in the regulation of mechanical load-induced growth. J Physiol 589: 5485-5501.

6. Faul F, Erdfelder E, Buchner A, Lang AG (2009) Statistical power analyses using G*Power 3.1: tests for correlation and regression analyses. Behav Res Methods 41: 1149-1160.

7. Okumura S, Kawabe J, Yatani A, Takagi G, Lee MC, et al. (2003) Type 5 adenylyl cyclase disruption alters not only sympathetic but also parasympathetic and calcium-mediated cardiac regulation. Circ Res 93: 364-371.

8. Okumura S, Takagi G, Kawabe J, Yang G, Lee MC, et al. (2003) Disruption of type 5 adenylyl cyclase gene preserves cardiac function against pressure overload. Proc Natl Acad Sci U S A 100: 9986-9990.

9. Okumura S, Vatner DE, Kurotani R, Bai Y, Gao S, et al. (2007) Disruption of type 5 adenylyl cyclase enhances desensitization of cyclic adenosine monophosphate signal and increases Akt signal with chronic catecholamine stress. Circulation 116: 1776-1783.

10. Kawasaki H, Springett GM, Mochizuki N, Toki S, Nakaya M, et al. (1998) A family of cAMP-binding proteins that directly activate Rap1. Science 282: 2275-2279.

11. de Rooij J, Zwartkruis FJ, Verheijen MH, Cool RH, Nijman SM, et al. (1998) Epac is a Rap1 guanine-nucleotide-exchange factor directly activated by cyclic AMP. Nature 396: 474-477.

12. Bristow MR (2000) beta-adrenergic receptor blockade in chronic heart failure. Circulation 101: 558-569.

13. Antonova L, Aronson K, Mueller CR (2011) Stress and breast cancer: from epidemiology to molecular biology. Breast Cancer Res 13: 208.

14. Shimizu Y, Khan M, Kato G, Aoki K (2018) Occlusal disharmony-induced stress causes osteopenia of the lumbar vertebrae and long bones in mice. Sci Rep 8: 173.

15. Yoshihara T, Matsumoto Y, Ogura T (2001) Occlusal disharmony affects plasma corticosterone and hypothalamic noradrenaline release in rats. J Dent Res 80: 2089-2092.

16. Katayama T, Mori D, Miyake H, Fujiwara S, Ono Y, et al. (2012) Effect of bite-raised condition on the hippocampal cholinergic system of aged SAMP8 mice. Neurosci Lett 520: 77-81.

17. Umeki D, Ohnuki Y, Mototani Y, Shiozawa K, Fujita T, et al. (2013) Effects of chronic Akt/mTOR inhibition by rapamycin on mechanical overload-induced hypertrophy and myosin heavy chain transition in masseter muscle. J Pharmacol Sci 122: 278-288.

18. Kilkenny C, Parsons N, Kadyszewski E, Festing MF, Cuthill IC, et al. (2009) Survey of the quality of experimental design, statistical analysis and reporting of research using animals. PLoS One 4: e7824.

19. National Research Council (US) Committee for the Update of the Guide for the C, Use of Laboratory Animals. Guide for the Care and Use of Laboratory Animals. Washington (DC): National Academies Press (US). 2011

National Academy of Sciences.

20. Miyata M, Suzuki S, Misaka T, Shishido T, Saitoh S, et al. (2013) Senescence marker protein 30 has a cardio-protective role in doxorubicin-induced cardiac dysfunction. PLoS One 8: e79093.

21. Yamamoto M, Yang G, Hong C, Liu J, Holle E, et al. (2003) Inhibition of endogenous thioredoxin in the heart increases oxidative stress and cardiac hypertrophy. J Clin Invest 112: 1395-1406.

22. Cohen J (1992) A power primer. Psychol Bull 112: 155-159.

23. Jin H, Fujita T, Jin M, Kurotani R, Namekata I, et al. (2017) Cardiac overexpression of Epac1 in transgenic mice rescues lipopolysaccharide-induced cardiac dysfunction and inhibits Jak-STAT pathway. J Mol Cell Cardiol 108: 170-180.

24. Ito A, Ohnuki Y, Suita K, Ishikawa M, Mototani Y, et al. (2019) Role of beta-adrenergic signaling in masseter muscle. PLoS One 14: e0215539.

25. Umeki D, Ohnuki Y, Mototani Y, Shiozawa K, Suita K, et al. (2015) Protective effects of clenbuterol against dexamethasone-induced masseter muscle atrophy and myosin heavy chain transition. PLoS One 10: e0128263.

---

## [Decision Letter · Decision Letter 1]

23 Jun 2020

PONE-D-19-25337R1

Effects of occlusal disharmony on cardiac homeostasis in mice

PLOS ONE

Dear Dr. Okumura,

Thank you for submitting your manuscript to PLOS ONE. After careful consideration, we feel that it has merit but does not fully meet PLOS ONE’s publication criteria as it currently stands. Therefore, we invite you to submit a revised version of the manuscript that addresses the points raised during the review process.

Importantly, this editor, who participated as a reviewer for the initial evaluation of this manuscript, found a critical issue regarding duplicate submission.

A concept of the study has been already described in an original article from the same authors (Yagisawa et al. Circ Cont. 2020). Moreover, parts of western blotting results, especially some bands for control and BO groups seem to be a part of the same membrane shown in the previously published article. (ratio of CaMKII is also weird. ~1200%(previous) vs ~300%(current) in BO group.) I understand that the published article was written in non-English language, however the article has an English abstract and is open-access so anyone can see the figures.

Although the reviewer is favorable to accept the manuscript after some minor revisions (see below), I strongly recommend the authors to disclose the use of previously published data in the revised manuscript (this probably may not fit to the PLOS ONE’s publication criteria), or significantly change the manuscript with great care for handling raw data, at least without showing the membrane and images, which were already presented elsewhere.

I would be willing to reconsider this manuscript after it has undergone a major revision.

We look forward to receiving your revised manuscript.

Kind regards,

Takashi Sonobe, Ph.D.

Academic Editor

PLOS ONE

Reviewers' comments:

Reviewer's Responses to Questions

**Comments to the Author**

1. If the authors have adequately addressed your comments raised in a previous round of review and you feel that this manuscript is now acceptable for publication, you may indicate that here to bypass the “Comments to the Author” section, enter your conflict of interest statement in the “Confidential to Editor” section, and submit your "Accept" recommendation.

Reviewer #2: All comments have been addressed

2. Is the manuscript technically sound, and do the data support the conclusions?

Reviewer #2: Yes

3. Has the statistical analysis been performed appropriately and rigorously? 

Reviewer #2: Yes

4. Have the authors made all data underlying the findings in their manuscript fully available?

Reviewer #2: Yes

5. Is the manuscript presented in an intelligible fashion and written in standard English?

Reviewer #2: Yes

6. Review Comments to the Author

Reviewer #2: The paper was overall improved. However, there are some issues. The paper should be revised.

TITLE

The authors used homeostasis, but it is not appropriate and not concrete. For example, “Effects of occlusal disharmony on cardiac fibrosis, myocyte apoptosis and myocyte oxidative DNA damage in mice”.

Please add the timing of serum corticosterone measurements and/or collection. The collection should be in the morning.

7. PLOS authors have the option to publish the peer review history of their article (what does this mean?). If published, this will include your full peer review and any attached files.

Reviewer #2: No

---

## [Author Response · Author response to Decision Letter 1]

7 Jul 2020

Reviewer #2: The paper was overall improved. However, there are some issues. The paper should be revised.

TITLE

The papers used homeostasis, but it is not appropriate and not concrete. For example, “Effects of occlusal disharmony on cardiac fibrosis, myocyte apoptosis and myocyte oxidative DNA damage in mice”.

Response: 

We modified the title as the reviewer suggested.

Please add the timing of serum corticosterone measurements and/or collection. The collection should be in the morning.

Response: 

We added the following sentences in the revised manuscript with a new reference (Page 8, Lines 9-10).

Blood sampling was done in the morning (9:00-10:00AM) and the procedure was completed within 30 s from the time of contact with the mouse [1]. 

Editor: --- I strongly recommend the authors to ----disclose the use of previously published data in the manuscript---, or significantly change the manuscript with great care for handling raw data, at least without showing the membrane and images, ---

Response: 

We sincerely apologize for omitting to mention our previous paper written in Japanese (Circ Cont 2020). Although we had described the BO-promoted increase of Bax/Bcl-2 ratio, CaMKII phosphorylation (Thr-286) and PLN phosphorylation (Ser-16, Thr-17) in that paper (Circ Cont 2020), it did not describe the effects of propranolol on the BO-promoted increase of these molecules, which are critical for the present manuscript. Accordingly, we adopted your second suggestion to avoid duplication, and have modified Fig 4A, Fig 4B, Fig 4D, Fig 5A and Fig 5B in the original version and incorporated the modified figures in the revised manuscript as Fig S6, Fig S8 and Fig S9. Thus, the revised manuscript no longer duplicates material from the previous paper. We apologize for having failed to spot this before. 

We also modified parts in the results section as shown below.

1) Page 23, Lines 10-16

Expression of Bax, an accelerator of apoptosis, in the heart was significantly increased by BO treatment --- in accordance with the previous study (Fig S6A) [2]. Propranolol alone had no effect on Bax expression, but blocked the BO-induced increase--- (Fig S6A).

2) Page 23, Line 17-Page 24, Line 5

We also found that the expression of Bcl-2, a decelerator of apoptosis, in cardiac muscle was significantly decreased by BO treatment (Control (n = 4) vs. BO (n = 4); 100 ± 19 vs. 55 ± 22 %, P = 3.5 x 10-2 by one-way ANOVA followed by the Tukey-Kramer post hoc test) in accordance with the previous study (Fig S6B) [2]. Propranolol alone had no effect on the Bcl-2 expression, but blocked the BO-induced decrease --- (Fig S6B).

3) Page 25, Lines 6-12

We thus examined the amounts of phospho-CaMKII (Thr-286) in the heart of BO mice and found that it was significantly increased --- in accordance with the previous study (Fig S8) [2]. Propranolol alone had no effect on the amounts of phospho-CaMKII (Thr-286), but propranolol blocked this increase --- (Fig S8). 

4) Page 26, Lines 3-12

Phospho-PLN (Thr-17) and phospho-PLN (Ser-16) were significantly increased in cardiac muscle of BO mice--- in accordance with the previous study (Fig S9A and S9B) [2]. Propranolol alone had no effect on the amounts of phospho-PLN (Thr-17 and Ser-16), but propranolol blocked both phosphorylations--- (Fig S9A and S9B).

References

1. Grootendorst J, Oitzl MS, Dalm S, Enthoven L, Schachner M, et al. (2001) Stress alleviates reduced expression of cell adhesion molecules (NCAM, L1), and deficits in learning and corticosterone regulation of apolipoprotein E knockout mice. Eur J Neurosci 14: 1505-1514.

2. Yagisawa Y, Suita K, Ohnuki Y, Ito A, Umeki D, et al. (2020) Effects of experimental malocclusion on cardiac function inmice. Circ Cont 41: 38-45.

---

## [Editor Report · Decision Letter 2]

10 Jul 2020

Effects of occlusal disharmony on cardiac fibrosis, myocyte apoptosis and myocyte oxidative DNA damage in mice

PONE-D-19-25337R2

Dear Dr. Okumura,

We’re pleased to inform you that your manuscript has been judged scientifically suitable for publication and will be formally accepted for publication once it meets all outstanding technical requirements.

Kind regards,

Takashi Sonobe, Ph.D.

Guest Editor

PLOS ONE
---

## [Editor Report · Acceptance letter]

14 Jul 2020

PONE-D-19-25337R2 

Effects of occlusal disharmony on cardiac fibrosis, myocyte apoptosis and myocyte oxidative DNA damage in mice 

Dear Dr. Okumura:

I'm pleased to inform you that your manuscript has been deemed suitable for publication in PLOS ONE. Congratulations! Your manuscript is now with our production department. 

Kind regards, 

on behalf of

Dr. Takashi Sonobe 

Guest Editor

PLOS ONE